# Predicting the impact of spatial heterogeneity on microbially mediated nutrient cycling in the subsurface

Swamini Khurana[1], Falk Heße[2,3], Anke Hildebrandt[2,4,5], Martin Thullner[1]

[1]Department of Environmental Microbiology, Helmholtz Centre for Environmental Research – UFZ, Leipzig, 04318, Germany
[2]Department of Computational Hydrosystems, Helmholtz Centre for Environmental Research – UFZ, Leipzig, 04318, Germany
[3]Institute of Earth and Environmental Sciences, University Potsdam, Potsdam, Germany
[4]Institute of Geoscience, Friedrich-Schiller-University Jena, Jena, Germany
[5]German Centre for Integrative Biodiversity Research, Leipzig, Germany

**Correspondence**: Swamini Khurana (swamini.khurana@gmail.com)

**Abstract.**

The subsurface is a temporally dynamic and spatially heterogeneous compartment of the Earth's Critical Zone, and biogeochemical transformations taking place in this compartment are crucial for the cycling of nutrients. The impact of spatial heterogeneity on such microbially mediated nutrient cycling is not well known which imposes a severe challenge in the prediction of in situ biogeochemical transformation rates and further of nutrient loading contributed by the groundwater to the surface water bodies. Therefore, we used a numerical modelling approach to evaluate the sensitivity of groundwater microbial biomass distribution and nutrient cycling to spatial heterogeneity in different scenarios accounting for various residence times. The model results gave us an insight into domain characteristics with respect to presence of oxic niches in predominantly anoxic zones and vice versa depending on the extent of spatial heterogeneity and the flow regime. The obtained results show that microbial abundance, distribution, and activity are sensitive to the applied flow regime and that the mobile (i.e., observable by groundwater sampling) fraction of microbial biomass is a varying, yet only a small, fraction of the total biomass in a domain. Furthermore, spatial heterogeneity resulted in anaerobic niches in the domain and shifts of microbial biomass between active and inactive states. The lack of consideration of spatial heterogeneity, thus, can result in inaccurate estimation of microbial activity. In most cases this leads to an overestimation of nutrient removal (up to twice the actual amount) along a flow path. We conclude that the governing factors for evaluating this are the residence time of solutes and the Damköhler number (Da) of the biogeochemical reactions in the domain. We propose a relationship to scale the impact of spatial heterogeneity on nutrient removal governed by the $\log_{10}$Da. This relationship may be applied in upscaled descriptions of microbially mediated nutrient cycling dynamics in the subsurface thereby resulting in more accurate predictions of e.g., carbon and nitrogen cycling in groundwater over long periods at the catchment scale.

## 1 Introduction

The Earth's Critical Zone comprises the near surface, surface and sub-surface compartments, from the top of the vegetation canopy to aquifers in the bedrock (Giardino and Houser, 2015;Küsel et al., 2016). Biogeochemical processes impact most ecosystem functions (and consequently ecosystem services) in the Critical Zone by controlling the distribution of nutrients in the compartments of the Critical Zone. All these compartments are connected by water fluxes. Within the Critical Zone, the soil and deeper subsurface compartments account for almost 50% of the global carbon budget, and the subsurface is also one of the biggest storage compartments of nitrogen (McMahon and Parnell, 2014;Schlesinger and Andrews, 2000). Especially the subsurface part of the Critical Zone exhibits high spatial and temporal variability in environmental conditions that have been proven to be correlated with subsurface nutrient dynamics (Cole et al., 2007;Harden et al., 1997;Holt, 2000;Küsel et al., 2016;van Leeuwen, 2000). Since studies investigating these links are limited to near-surface soil zones, e.g. focusing on the root zone (Küsel et al., 2016), studies linking surficial events with nutrient dynamics in the deeper subsurface are limited. Some research however shows that both, subsurface heterogeneity and input variation, affect subsurface microbial community structure. Schwab et al. (2017);Zhou et al. (2012);Hofmann et al. (2020) linked changing diversity of microbial communities in groundwater with spatio-temporal variation of the groundwater physico-chemical quality. McGuire et al. (2000) and Benk et al. (2019) linked changing composition of terminal electron acceptors and of dissolved organic matter (DOM) in groundwater with surficial events, respectively. Their results also indicated further links with microbial community evolution, but they were unable to resolve the effect of transport in the subsurface presumably due to unresolved spatial heterogeneity. All the aforementioned studies combined establish a link between spatial-temporal variability in environmental conditions and nutrient cycling. However, this link is not yet quantitatively characterized. Therefore, this further impedes the predictability of biogeochemical cycles.

Improved prediction of biogeochemical cycles requires advancement in mechanistic understanding of governing factors. Microbial communities play a key role in these biogeochemical cycles since they mediate nearly all the naturally occurring processes that contribute to these cycles. Recent advances in microbial techniques have led to greater insight into the functions of microbial communities for biogeochemical transformations in laboratory scale batch and column experiments (Ballarini et al., 2014;Grösbacher et al., 2018). However, transferring this knowledge to the subsurface is challenging. For instance, the growth conditions used in laboratory studies are favourable with high substrate concentrations and readily accessible terminal electron acceptors (Grösbacher et al., 2018;Hofmann and Griebler, 2018). This is not representative of the subsurface as the subsurface is a spatially heterogeneous medium. Spatial heterogeneity influences subsurface microbial and nutrient dynamics by limiting access to nutrients and electron acceptors (Murphy et al., 1997), thereby influencing the distribution of active, inactive, suspended

and attached microbes as well (Grösbacher et al., 2018; Couradeau et al. 2019). Inactive microbes were found to account for 60% to 80% of total microbial biomass in soil (Lennon and Jones, 2011), and attached microbes commonly form the majority of microbial biomass fraction in the subsurface (Griebler and Lueders, 2009; Grösbacher et al., 2018). However, data on these fractions for groundwater systems are still scarce. Investigating the impact of spatially heterogeneous media on microbial biomass and nutrient cycling in the subsurface is hindered by the limited observational opportunities, lack of visualization of real time conditions and limitations of sampling methods and oligotrophic conditions (growth limiting) in groundwater (Ballarini et al., 2014;Hofmann and Griebler, 2018). Since the Critical Zone is a complex system with non-linear process dynamics, governing factors are difficult to isolate, and their impact is unfeasible to quantify (Grösbacher et al., 2018). To overcome these limitations, numerical modelling approaches are powerful alternatives to undertake such investigations (Molins et al., 2014).

Formulating a conceptual model for microbially mediated carbon and nitrogen dynamics in the subsurface requires a two-pronged approach. First, the reaction network should be representative of a system's chemical and biological species, and second, the flow component of the model representative of a system's flow and transport pathways. Biogeochemical reaction networks have been explored extensively over the past decades with improvement in the conceptual understanding of the transient environmental conditions of the Critical Zone, the microbial life cycle, and the key processes involved in carbon and nitrogen cycles (Thullner and Regnier, 2019; Manzoni and Porporato, 2009). Incorporating microbially explicit reaction networks in reactive transport models is beneficial as these models could capture transient conditions and associated impacts (Thullner et al., 2007). In contrast to soil-based models that account for complex reaction networks, often comprising more than one microbial functional group (Yabusaki et al., 2017b;Thullner et al., 2007;Thullner and Regnier, 2019;Manzoni and Porporato, 2009), the reaction networks used for modelling biogeochemical processes in deep sub-surface domains are seldom complex. They do not account for microbially explicit models and relevant microbial life processes or any interactions thereof (Thullner and Regnier, 2019). A straight-forward application of the soil-based biogeochemical model approaches to conditions in deeper subsurface compartments is problematic because the nature of carbon source changes as it travels into the deeper zones. A reaction network that is sufficiently representative of growth conditions found in the subsurface is lacking and must be conceptualized to study both microbial dynamics and resulting nutrient dynamics. Below we present a possible reaction network for such groundwater settings.

The second challenge, as stated above, is to characterize the flow and transport in a heterogeneous medium. Several attempts have already been made to model microbially driven reactions in the subsurface (Yabusaki et al., 2017b;Thullner et al., 2005;Schäfer et al., 1998a;Hunter et al., 1998;Arora et al., 2016) at a regional scale with further investigations on the impact of temporal variation on microbial activity and microbially-

driven redox dynamics in riparian zones (Yabusaki et al., 2017a;Dwivedi et al., 2018;Arora et al., 2016). Conducting studies at this scale is relevant but it lacks spatial resolution of microbially mediated nutrient dynamics in the subsurface. Additionally, it is difficult to transfer the results to other geological settings (Tufenkji, 2007).

To understand the fundamental mechanisms (without the volume averaging effect of large-scale studies) influencing microbial activity, several studies worked on identifying factors influencing microbial activity at the pore scale (Stolpovsky et al., 2011;Meile and Tuncay, 2006;Heße et al., 2010; King et al., 2010; Gharasoo et al., 2012). Exploring microbial dynamics at the pore scale requires the knowledge of pore scale features/geometry for practical applications (Heße et al., 2010), which is typically not available. Additionally, utilizing the pore scale resolution as the base for modelling catchment scale nutrient cycles is computationally problematic. Meanwhile, field groundwater sampling techniques reflect average conditions at the continuum scale depending on the sampling resolution. Sanz-Prat et al. (2016) attempted to simplify reactive transport modelling in heterogeneous media at the meter scale by proposing a travel time approach but considered a limited reaction network comprising only growth and decay dynamics of aerobic degraders and denitrifiers. The study conducted by Jung and Meile (2019) applied first-order reactions in heterogeneous porous media at the Darcy scale (or continuum scale) and further upscaled the effective reactions to the regional scale. Microbial kinetics and interplay between different functional groups thereof are more accurately expressed using Monod derived kinetics (Arora et al., 2016;Thullner et al., 2007) although Liu et al. (2019) attempted to identify the conditions in which first-order rates may be suitably used in soil systems to optimize computational efforts at field or regional scales. In summary, model attempts have been related to the regional and pore scale, leaving a gap at the soil core, rock core and groundwater sampling scale.

In this research, we aim to study nutrient dynamics using a comprehensive reaction network at the continuum scale (sub-meter scale in our case). This provides the link between the pore scale microbial dynamics and regional scale microbial dynamics. It assists in developing a process-based understanding of the impact of spatial heterogeneity on microbial activity and subsequent nutrient dynamics and assists in scaling the activity to pragmatic regional scales accounting for spatial heterogeneity.

We seek to describe the influence of spatial variability of terrestrial subsurface settings (i.e., porous aquifer properties) on the in situ biogeochemical function of microorganisms through numerical simulations. Since preferential flow paths have been established to control access to nutrients, electron acceptors and thus influence the emergence of microbial hotspots (Franklin et al., 2019), we focus on investigating spatial heterogeneity alone. We use a complex reaction network that considers varying microbial functional groups (both aerobes and anaerobes), key microbial life processes in a variety of redox conditions (aerobic, ammonia oxidizing, nitrate reducing and sulphate reducing) eventually influencing carbon and nitrogen

transformation. Simulated scenarios are informed by data from the literature and from a subject site to describe realistic although generic conditions, which allows us to combine these conditions with different types of subsurface heterogeneities to determine the resulting biogeochemical potential of the subsurface 140 system. The results of this study support the identification of key drivers of microbial dynamics in the Critical Zone and assist in effective upscaling of these process descriptions. This, in turn, contributes towards the regional scale modelling of biogeochemical cycles resulting from microbial dynamics.

## 2 Methods

This study investigates the impact of spatial heterogeneity of the aquifer matrix on nutrient cycling in 145 groundwater with a focus on carbon and nitrogen using reactive transport modelling. For this we used a numerical reactive transport modelling approach which considered the microbial abundance and activity in spatially heterogeneous environmental conditions, that is, spatial variations of aquifer permeability. We used the geochemical and geomicrobial observations from our subject site in the Hainich Critical Zone Exploratory (CZE, Küsel et al. (2016)) as the foundation of the conceptual model to investigate the research 150 questions. The subject site was set up under the DFG Collaborative Research Centre Grant 1076 AquaDiva to study the links between surficial processes and subsurface dynamics. Thus, it provides spatially and temporally resolved field observations to enable the formulation of a representative conceptual model. We used this information to constrain our conceptual approach and the simulated scenarios to realistic conditions. It is however not the aim to explicitly simulate a specific part of the subject site. For some model 155 input we rather considered values at the extreme end of possible conditions to enlarge the range of conditions covered by our model scenarios. We ran all simulations for a two-dimensional transect of 50 x 30 cm size assuming fully saturated conditions, steady-state flow and constant inflow concentrations of dissolved species. We deemed this domain size appropriate to investigate sub-sampling (sub-meter) scale heterogeneities. We considered three different average flow velocities and 12 scenarios of hydraulic 160 conductivity fields of varying heterogeneity for all simulations. The following sections describe the conceptual model comprising reaction network, flow regime and corresponding parameterization, the simulated scenarios and methodology of analyses of the simulation results.

### 2.1 Reaction Network

We conceptualized an extended biogeochemical process network to describe the turnover of carbon and 165 nitrogen (Appendix A and Fig. 1). The reaction network is an extended adaptation of the carbon dynamics described by Vogel et al. (2018), and relevant processes in the subsurface as implemented by Manzoni and Porporato (2009). The network accounts for autotrophy, and heterotrophy in both aerobic and anaerobic

regimes considering four functional groups of microorganisms: aerobic dissolved organic carbon (DOC) degraders, nitrate reducing DOC degraders, ammonium oxidizers and sulphate reducing DOC degraders.

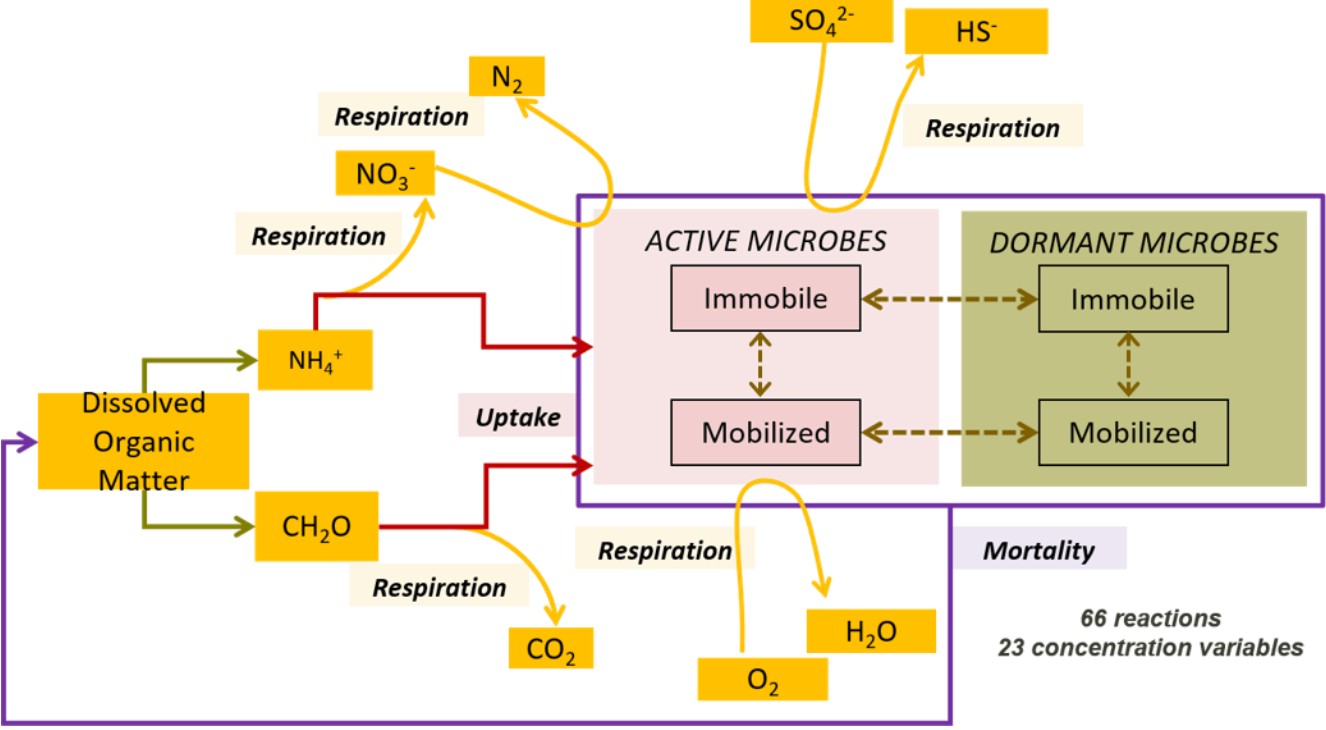

**Figure 1: Schematic representation of the simulated biochemical reaction network**

The network accounts for other observed microbial processes such as dormancy and mortality using a modified dual-Monod approach adapted from Stolpovsky et al. (2011). The reaction network also accounts for the "maximum carrying capacity" of the matrix (Ding, 2009;Grösbacher et al., 2018), lumping all
growth-limiting effects not explicitly accounted for into an additional term (Prommer et al., 1999;Schäfer et al., 1998b;Thullner et al., 2007;Wirtz, 2003). Eventually the carbon and nitrogen loop are completed via recycling of bacterial necromass. Furthermore, the reaction network accounts for microbial attachment, in case of hospitable conditions, and detachment due to inhospitable conditions or velocity of the water (see section A.3.3). The detached mobile bacteria are transported by the flowing water.

**2.2 Flow and Transport**

We modelled steady-state flow conditions in each fully saturated domain (50 x 30 cm in size) by imposing fixed hydraulic heads at the inlet and outlet of the domain adjusting the inlet value to achieve the desired average flow velocity. We kept the head at both, inlet and outlet, constant throughout the simulation periods ensuring steady-state flow conditions. All simulated domains had a constant porosity of 0.2 and an average
hydraulic conductivity of $2.0 \ 10^{-6}$ m s$^{-1}$. The transport regimes account for advection, dispersion and diffusion. We assumed inlet concentrations of mobile species to be constant for all simulations.

## 2.3 Parameterization

The subject site is a monitoring well transect within the Hainich CZE in the Hainich National Park, Thuringia, Germany. Groundwater characteristics and composition of microbial communities observed in the groundwater of the subject site over five years (Küsel et al., 2016) informed the parameterization of the model.

As model input, we introduced a solution which was representative of water infiltrating from the shallow subsurface (Table A3), containing a mixture of naturally derived dissolved organic carbon (DOC), dissolved oxygen (DO), nitrate, sulphate and some mobile microorganisms (heterotrophic aerobic degraders, heterotrophic nitrate and sulphate reducers and autotrophic ammonia oxidizers) in the domain. The concentrations of the reactive species mimicked conditions observed in the subject site.

## 2.4 Simulated Scenarios

We performed simulations for three different flow regimes, each characterized by a specific average flow velocity: for the slow flow regime, the average flow velocity of $3.8 \times 10^{-4}$ m d$^{-1}$ is given by the estimated recharge rate at the subject site (Kohlhepp et al., 2017;Jing et al., 2018) and represents the slow migration of water through the uppermost part of the saturated aquifer. We increased the average flow velocity by a factor of 10 for the medium flow regime, and by a factor of 100 for the fast flow regime (Table 1).

**Table 1: Flow and transport parameters considered in the simulations and the resulting Peclet number (Pe) associated with the different flow regimes. For the latter, the domain size of 0.5 m was used as characteristic length for all flow regimes.**

| Property | Slow flow | Medium flow | Fast flow |
|---|---|---|---|
| Darcy velocity (m d$^{-1}$) | $3.8 \times 10^{-4}$ | $3.8 \times 10^{-3}$ | $3.8 \times 10^{-2}$ |
| Diffusion coefficient (m$^2$ d$^{-1}$) | $8.64 \times 10^{-5}$ | $8.64 \times 10^{-5}$ | $8.64 \times 10^{-5}$ |
| Longitudinal dispersivity (m) | 0.02 | 0.02 | 0.02 |
| Pe (-) | 2.02 | 11.7 | 22.45 |

For each flow regime, a base case scenario accounted for a homogeneous flow field, i.e., the homogeneous domains did not have any variation in the distribution of conductivity field, and no associated anisotropy. Further scenarios considered spatial heterogeneity of the flow field using randomly generated hydraulic conductivity fields (Heße et al., 2014). Each random field was characterized by the same mean value of conductivity (i.e., average conditions at the subject site (Jing et al., 2018)) and spatial autocorrelation length scale (0.1 m) in all realizations, scaling with the size of the domain in line with previous studies (Turcke

and Kuper, 1996; Welhan and Reed, 1997; Desbarats and Bachu, 1994). To conceptualize heterogeneity, we used a limited parameter set, variance in the log normal distribution of conductivity and anisotropy, to

represent varying porous and fractured media and to also control the degree of channelized flow in the domain (Edery et al., 2016; Heße et al., 2014). We varied the values of these parameters within ranges reflecting the site conditions/geological features at the study site (Heath, 1983;Kohlhepp et al., 2017). The scenarios are summarized in Table 2. In total, we ran 147 simulations for the three different flow regimes in spatially heterogeneous domains. For each scenario we kept the average water fluxes the same in all

scenarios, and we compared the results of the scenarios with the base case scenario. We used the breakthrough of a constantly injected conservative tracer as a measure of the solute residence time (i.e., time for flux averaged outlet concentration to reach 50% of inlet value) in the system.

### 2.5 Numerical Tools

We used OGS#BRNS (Centler et al., 2010) to carry out the numerical simulations. This numerical model

couples the BRNS (Biochemical Reaction Network Solver (Aguilera et al., 2005;Regnier et al., 2002)), an established tool that allows for the simulation reaction networks of arbitrary size and complexity (Thullner et al., 2005) with OGS (Open Geosys), a state of the art open source thermo-hydro-mechanical-chemical (THMC) simulator (Kolditz et al., 2012) that has also been used for modelling groundwater flow and transport (Jing et al., 2018). We used a constant finite volume discretization of 0.01 m in both directions.

Transient simulations were performed until steady state was achieved.

**Table 2: Summary of spatially heterogeneous scenarios investigated for each flow regime. S. No. 1 is the homogeneous base case.**

| S. No. | Variance in permeability | Anisotropy | Number of realizations | Category type |
|--------|--------------------------|------------|------------------------|---------------|
| 1 | 0 | Not applicable | 1 | Homogeneous (referred as 0:1 in Fig. 2) |
| 2 | 0.1 | 2 | 4 | 0.1:2 |
| 3 | 0.1 | 5 | 4 | 0.1:5 |
| 4 | 0.1 | 10 | 4 | 0.1:10 |
| 5 | 1 | 2 | 4 | 1:2 |
| 6 | 1 | 5 | 4 | 1:5 |
| 7 | 1 | 10 | 4 | 1:10 |
| 8 | 5 | 2 | 4 | 5:2 |
| 9 | 5 | 5 | 4 | 5:5 |

| S. No. | Variance in permeability | Anisotropy | Number of realizations | Category type |
|---|---|---|---|---|
| 10 | 5 | 10 | 4 | 5:10 |
| 11 | 10 | 2 | 4 | 10:2 |
| 12 | 10 | 5 | 4 | 10:5 |
| 13 | 10 | 10 | 4 | 10:10 |

We used the Python programming language (van Rossum and Drake, 2006) (referred to as Python henceforth) to set up the scenarios for running the simulations using OGS#BRNS. These tasks included the generation of input files. We used ogs5py (Müller, 2020) to generate the input files for running the simulations in OGS#BRNS. We used gstools (Müller and Schüler, 2019) to generate the spatial random fields to represent heterogeneous domains in OGS#BRNS. We processed and further analysed simulation
results using a workflow in Python as well. We also used Python to generate all graphical outputs presented in this paper. The scripts used for the Python workflow along with the input files are available in a repository for ease of reproducibility (Khurana et al., 2021).

**2.6 Data Analysis**

The Peclet number (Pe) indicates the relative importance of flow processes in the flow regime. The resulting
Pe of each flow regime (calculated using Eq.(1)) increased from 2 indicating a mixed diffusion-advection-transport regime for the slow flow regime to 22 indicating fully advection-dominated transport for the fast flow regime (see Table 1 for further details).

$$Pe = \frac{v_{eff} \cdot l}{D + \alpha \cdot v_{eff}}, \tag{1}$$

with $v_{eff}$ as effective Darcy velocity, l as length scale, D as diffusion coefficient, and α as longitudinal
dispersivity.

The breakthrough time is a useful metric to evaluate the matter flux in the domain. We defined the breakthrough time of a conservative tracer as the time taken for the flux averaged concentration at the outlet of the domain to be 50% of the continuous tracer input concentration at the inlet of the domain. This also enables evaluating impact of spatial heterogeneity on matter flux alone, without considering impact of
reactions.

To evaluate impact of spatial heterogeneity on nutrient cycling, we calculated removal of reactive species (that is, DOC, DO, ammonium, and nitrate) from the domain in steady state conditions. Thus, while the chemical species entering the domain at the inlet were consumed at varying rates by the microbial species present in the system, the rate of consumption was constant in time in each domain in all flow regimes. In

addition to these dissolved reactive species, we also considered (total) nitrogen and total organic carbon (TOC) concentrations by considering also nitrogen and carbon present in the mobile microbial biomass and in particulate organic matter being transported in the domain (Appendix A). We compared the changing mass removal in heterogeneous domains with the respective base case scenarios (homogeneous domains). To evaluate the key factors determining the impact of spatial heterogeneity on nutrient cycling, we

undertook a series of multivariate statistical analyses of the simulation results using Linear Mixed Effect Modelling, progressively including variables in both fixed effects and random effects. We compared the Akaike Information Criterion (AIC) of each model to evaluate the fit of the model. AIC is an indicator of prediction error associated with a general linear model. It is an indicator of relative performance of a group of models; the model with the lowest AIC is concluded to be the one with least prediction error or best

performance. With each iteration of the model, we selected the features most influencing the performance of the model and reducing the AIC of the predictions. We described these key factors using established dimensionless numbers which are also identifiable by observations. For example, we used Pe to indicate different flow regimes (described in Sect. 2.4). Similarly, we used the Damköhler number (Da) to indicate the reaction regime for each reactive species. Da is defined as the ratio of the transport time scale and the

reaction time scale as described in Eq. 2.

$$Da = \frac{\tau_{transport}}{\tau_{reaction}}, \tag{2}$$

where, $\tau_{reaction}$ is the characteristic reaction time scale and $\tau_{rtransport}$ is the characteristic transport time scale given by the breakthrough time of a conservative tracer in the domain. We adapted this definition to derive characteristic reaction time scale assuming 63% loss (Pittroff et al., 2017) and used Eq 3 below to calculate

the apparent Da using values estimable in the field when $\frac{C_{out}}{C_{in}} > 5\%$.

$$Da = -\ln\frac{C_{out}}{C_{in}}, \tag{3}$$

with $C_{in}$ as flux averaged concentration of a reactive species entering the domain, and $C_{out}$ as flux averaged concentration of the reactive species leaving the domain. In case $\frac{C_{out}}{C_{in}} \leq 5\%.$, we used Eq. 4 and Eq. 5 to derive the apparent Da of the chemical species

$$\tau_{reaction} = \frac{-\ln(0.37)}{-\ln(\frac{C_{y5}}{C_{in}})} \times \tau_{y5}, \tag{4}$$

$$\tau_{reaction} = \frac{\tau_{y5}}{\ln(\frac{C_{y5}}{C_{in}})}, \tag{5}$$

where, $C_{y5}$ is the concentration of the chemical species at the first cross-section (y = y5) when $\frac{C}{C_{in}} \leq 5\%$, and $\tau_{y5}$ is the breakthrough time for a conservative tracer at the same cross-section, i.e., y = y5. $\tau_{transport}$ in this case was the same as the breakthrough time of the conservative tracer in the domain (Eq. 6).

$$Da = \frac{breakthrough\ time}{\frac{\tau_{y5}}{\ln(\frac{Cy5}{Cin})}},$$ (6)

Thus, we were able to characterize reaction dominant systems where Da > 1. We took the logarithm of Da to the base 10 ($\log_{10}Da$) to characterize the regime for each reactive species in each domain.

For a scalable relationship addressing impact of spatial heterogeneity on reactive species removal, we conduct a simple linear regression analysis of species removal vs. residence time (both in relative units to

295 the homogeneous reference cases) for different $\log_{10}Da$ ranges.

For comparison we also use the following expression to predict the impact of reducing breakthrough time on removal of reactive species, in case of a first order removal rate expression (Eq. 7):

$$C_t = C_i e^{-kt},$$ (7)

with $C_i$ as initial concentration of reactive species [ML$^{-3}$], $C_t$ as concentration of reactive species at time t

[ML$^{-3}$], k as first order rate constant [T$^{-1}$], and t as time taken for the reaction to occur [T].

Then it follows that, normalized removal of reactive species may be described with:

$$\frac{C_i - C_t}{C_i} = 1 - e^{-kt}$$ (8)

To compare the removal of reactive species between two different time points, we use:

$$Impact\ on\ removal\ of\ reactive\ species\ with\ respect\ to\ base\ case = \frac{1 - e^{-Da.tf}}{1 - e^{-Da}}$$ (9)

with tf as ratio of the time taken for the reaction to take place in the two (2) different scenarios. In our study, this is the same as the ratio of breakthrough time in the heterogeneous domain and that in the base case. Furthermore, we calculated the impact of reducing breakthrough time on removal of reactive species, in case of a zeroth order (i.e. constant) removal rate $R_0$ as

$$Impact\ on\ removal\ of\ reactive\ species\ with\ respect\ to\ base\ case = tf\ R_0$$ (10)

**3 Results**

We compare characteristics of flow and transport of porous media such as conservative tracer breakthrough, microbial biomass in the domain and nutrient removal from the domain for heterogeneous domains and the base case. The base case is the homogeneous domain in all the three considered flow regimes. We explore flux-averaged concentrations of mobile species and spatially averaged concentrations of immobile species

in 1-D, along the predominant flow direction, and explore the 2-D concentration heat maps of the domain to compare the information lost when neglecting spatial heterogeneity at scales smaller than that of the sample. We further consider the total microbial biomass present in the domain, and nutrient removal from the domain as aggregated results and compare these between the heterogeneous domains and respective base cases.

## 3.1 Base case (homogeneous domain with uniform flow rate) results

The breakthrough time varied in the base case of each flow regime depending on the flow velocity in the domain. It was 205 days in the slow flow regime, 24 days in the medium flow regime and 2.4 days in the fast flow regime.

As mentioned in Sect. 2.3, we set the concentration of the dissolved species at the inlet to be the same across all flow regimes and heterogeneity scenarios, while it varied at the outlet for each scenario. In all flow regimes DOC concentrations decreased continuously along the domain length, yet they remained at relatively high values. In other words, an active microbial DOC degradation in the entire domain was not significantly limited by the abundance of DOC itself. In the slow flow and medium flow regimes, the dissolved oxygen (DO) dropped to concentrations less than 3 µM (common detection limit of DO sensors (ISO, 2014)) within the top half (upgradient) of the domain, indicating anoxic conditions in the downgradient parts of the domain (Fig. S1). Along the 1-D flow path in the domains aerobic degradation rates decreased more and more at low concentrations of DO (below approximately 20 µM), while ammonia oxidation persisted. With DO concentration lowering further, nitrate concentration reduced which is attributable to the activity of nitrate reducers at DO < 15 µM (Fig. S1). As the concentration of DO reduced, so did the biomass of aerobic degraders, while ammonia oxidizer biomass increased. This resulted in preferential occurrence of ammonia oxidation and nitrate reducers and nitrate reduction further downgradient in the domain (Fig. S2). No sulphate reduction took place in any of the flow regimes; the concentration of nitrate was still high (> 63 µM in all flow regimes) down to the outlet. In contrast to the slow and medium flow regime, DO concentration at the outlet of the fast flow regime (~4 µM in the base case) indicated that both oxic zones and aerobic activity prevailed further downgradient in the domain and consequently the growth of nitrate reducers was suppressed till further downgradient in the domain. Overall, the concentration profiles along the flow direction of the base case in all flow regimes were thus in agreement with redox hierarchy wherein aerobic degradation occurred preferentially upgradient in the domain promoted by a relatively high concentration of aerobic degraders.

The removal of reactive species, DOC (59.2%), DO (99.6%), ammonium (19.8%) and nitrate (74.7%), was the highest in the slow flow regime (Table 3). The removal of the reactive species was related to the average flow velocities since it related directly to the residence time in the domain and reaction dominated regimes. Hence, the rate of removal of all these reactive species reduced in medium flow and fast flow regimes. Also the removal of total nitrogen was the highest in the slow flow regime (57%), while the removal of TOC was the lowest there (32.6%) and highest in the medium flow regime (42.6%).

**Table 3: Removal of dissolved species ($R_b$) in terms of mass flux ($\dot{m}$ in µmol d$^{-1}$) from the homogeneous domain in three flow regimes - slow flow, medium flow and fast flow**

| Dissolved Species | Slow flow | | | Medium flow | | | Fast flow | | |
|---|---|---|---|---|---|---|---|---|---|
| | $\dot{m}_{in}$ | $\dot{m}_{out}$ | $R_b(\%)$ | $\dot{m}_{in}$ | $\dot{m}_{out}$ | $R_b(\%)$ | $\dot{m}_{in}$ | $\dot{m}_{out}$ | $R_b(\%)$ |
| DOC | 0.456 | 0.186 | 59.2 | 4.56 | 1.98 | 56.5 | 45.6 | 31.4 | 31.1 |
| DO | 0.143 | 0.001 | 99.6 | 1.43 | 0.01 | 99.4 | 14.3 | 0.2 | 98.4 |
| Ammonium | 0.0342 | 0.0274 | 19.8 | 0.342 | 0.276 | 19.4 | 3.42 | 3.03 | 11.5 |
| Nitrate | 0.143 | 0.036 | 74.7 | 1.43 | 0.49 | 65.8 | 14.3 | 14.1 | 1.12 |
| Nitrogen | 0.178 | 0.077 | 57.0 | 1.78 | 0.84 | 53.1 | 17.9 | 17.6 | 1.90 |
| TOC | 0.470 | 0.316 | 32.6 | 4.70 | 2.70 | 42.6 | 47.0 | 36.3 | 22.7 |

The concentration of microbial species in different states of activity and locations in the domain is shown in Table 4. The total biomass concentration was the highest in the slow flow regime (122 µM C), while it was the lowest in the fast flow regime (86 µM C). This reduction was mainly attributed to a decrease in mobile biomass concentration with increasing flow rate while the total concentration of immobile biomass remained constant with changing flow regimes. In all the flow regimes, the aerobic degraders formed the

dominant species, primarily due to the influx of oxygenated water at nearly saturation levels entering the domain at the inlet. In the slow flow regime, the highest proportion of biomass was contributed by inactive microbial species (>90% of the total biomass concentration). The proportion of active aerobic degraders and ammonia oxidizers was the lowest in the slow flow regime (~5%) while it increased in the medium flow regime (~17%) and it was the dominating species in the fast flow regime (~87%). This was indicative

of a small oxic zone with aerobic activity in the slow flow regime domain, which further expanded downgradient in the medium flow regime domain (Fig. S1 and Fig. S2). The dominance of the active aerobic degraders and increased presence of ammonia oxidizers in the fast flow regime domain indicated persistent oxic conditions and aerobic activity. Consequently, the proportion of active nitrate reducers was lowest in the fast flow regime (~3%), only growing in the downgradient direction near the outlet of the domain (Fig.

S2). The medium flow regime provided the conditions for active nitrate reducers to sustain and form a substantial proportion of the microbial community (14% as opposed to ~4% in slow flow regime and ~3% in fast flow regime). Among the active microbial species, the immobile fraction was higher than the mobile fraction in all flow regimes (more than 7 times in the slow flow regime, more than 4 times in the medium flow regime and more than 2 times in the fast flow regime).

**Table 4: Total biomass concentration (µM C) in homogeneous base case domains (volume averages) with fraction of biomass concentration (%) of each microbial species for three flow regimes**

| Microbial Species | Slow flow | Medium flow | Fast flow |
|---|---|---|---|
| Total | 122.0 | 93.34 | 86.35 |
| Active fixed aerobes | 3.5 | 12 | 74 |
| Active fixed ammonia oxidizers | 0.5 | 2.5 | 2.4 |
| Active fixed nitrate reducers | 2.5 | 12 | 2.3 |
| Active mobile aerobes | 1.0 | 2.2 | 9.6 |
| Active mobile ammonia oxidizers | 0.2 | 0.7 | 0.7 |
| Active mobile nitrate reducers | 1.2 | 3.1 | 0.3 |
| Inactive fixed aerobes | 44 | 41 | 5.2 |
| Inactive fixed ammonia oxidizers | 0.5 | 0.2 | 0.2 |
| Inactive fixed nitrate reducers | 15 | 11 | 3.2 |
| Inactive mobile aerobes | 24 | 12 | 0.5 |
| Inactive mobile ammonia oxidizers | 0.3 | 0.1 | 0.3 |
| Inactive mobile nitrate reducers | 7.9 | 2.9 | 1.4 |

## 3.2 Tracer breakthrough times

For each flow regime, the tracer breakthrough time in heterogeneous domains varied from that in the base case. With increase in variance of the hydraulic conductivity field and increase in anisotropy in the domain, the breakthrough time was shorter compared to the base case (Fig. 2). This was a result of preferential flow paths that were introduced by the heterogeneous hydraulic conductivity fields. The same "category" (combination of variance and anisotropy) of heterogeneity induced varying impact depending on the flow regime, with higher average flow velocities leading to relatively stronger reductions of the breakthrough times. This difference in the impact of heterogeneity on tracer breakthrough times and thus the residence time of solutes in the domain was attributed to the different Peclet numbers (Pe) of the regimes (Table 1). Diffusion played a stronger role in the transport processes in the slow flow regime, promoting mixing effects and reduced influence of the preferential flow paths in heterogeneous domains. This resulted in the lower deviation in breakthrough time from the base case in the slow flow regime. In contrast, in the medium and in particular in the fast flow regime transport was dominated by advection with little mixing between flow paths. The preferential flow paths in the heterogeneous domains therefore had a higher influence on the resulting tracer breakthrough times, and thus on the residence time of dissolved species in these regimes.

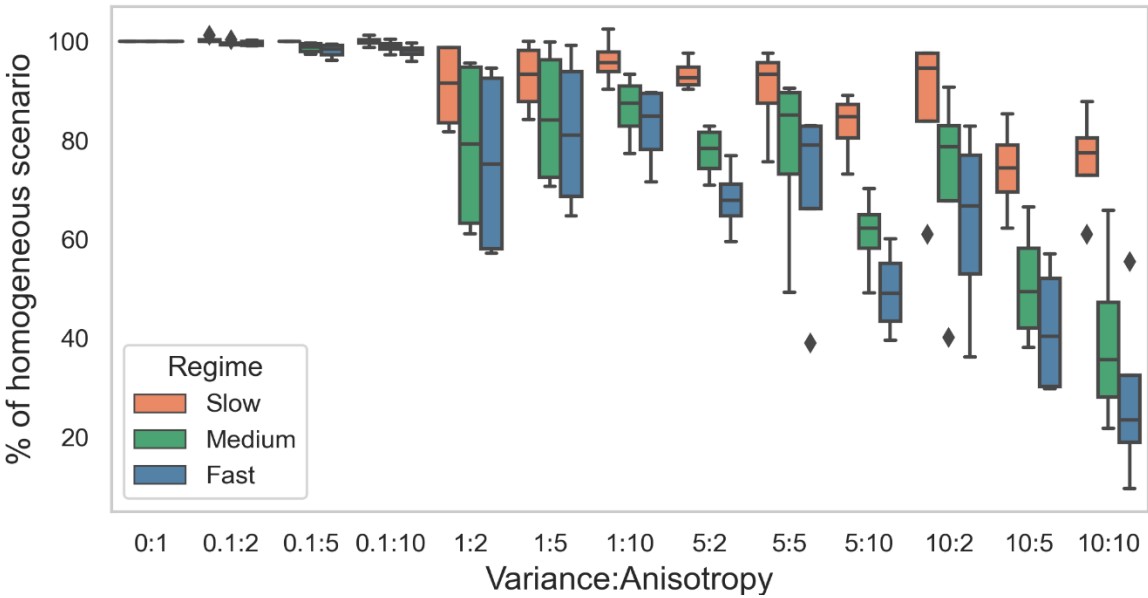

**Figure 2: Breakthrough time in different heterogeneous scenarios (described as variance in permeability field:anisotropy) normalized by that in the base case (or homogeneous case) in three flow regimes: Slow, medium and fast flow.**

### 3.3 Distribution of dissolved reactive species in heterogeneous scenarios

Scenarios with a heterogeneous hydraulic conductivity distribution exhibited a heterogeneous flow velocity distribution with pronounced preferential flow paths emerging with increasing variance and/or anisotropy of the conductivity distributions. The distribution of dissolved species in heterogeneous domains followed the orientation of the preferential flow paths (Fig. S3). All the species persisted longer along these preferential flow paths compared to the low permeability zones. Moreover, also on average all the reactive species penetrated further downgradient into the heterogeneous domains compared to the homogeneous domain due to the presence of the preferential flow paths (Fig. S1). For example, in the medium and fast flow regime DO persisted further in the heterogeneous domain (deeper in the domain) as the groundwater flowed through preferential flow paths. This impact of heterogeneity on longer persistence of DO was, however, not observable for the slow flow regimes. This is because the DO was preferentially and quickly consumed by aerobic degraders close to the inlet of the domain in the slow flow regime, rendering more than 90% of the domain sub-oxic to anoxic with prevailing anaerobic activity. Effectively, spatial heterogeneity did not play a role in aerobic respiration in the slow flow regime. In contrast, a larger oxic zone with aerobic activity existed in the upgradient section of the domains in medium and fast flow regimes. There, spatial heterogeneity resulted in observable shifts of the transition from oxic to sub-oxic zone or from aerobic activity to anaerobic activity to further downgradient parts of the domain. Additionally, spatial heterogeneity resulted in oxic and anoxic mesh nodes coexisting along a cross-section that was apparently

oxic (Fig. 3). Oxic mesh nodes were nodes where DO was recorded to be higher than 3 µM. We noted that even though the flux averaged concentration decreased steadily in the downgradient direction, a high percentage of nodes along the cross-section remained oxic in heterogeneous domains. Because of this delayed transition, nitrate reduction was also affected. In heterogeneous domains, nitrate was observed to be respired further downgradient in the domain and at the interface of high flow and low flow zones (Fig. S3).

These concentration distributions translated into reduced removal of carbon and nitrogen in heterogeneous domains with increasing spatial heterogeneity compared to the base cases (Fig. 4). DOC removal was less than in the base case in all the flow regimes with lowest removal reaching only 40% of the base case values in the fast flow regime. The removal of DO was reduced in the fast flow regime (down to 40% of the base case value) while no or negligible reductions were observed for most slow and medium flow scenarios. Nitrogen removal was reduced in the slow and medium flow regimes yet reaching at least 70% of base case values. One exception was nitrogen removal in the fast flow regime, which increased (up to 6 times the base values) compared to the base case. The dependency of TOC removal on spatial heterogeneity matched that of DOC for the different flow regimes (Fig. 4).

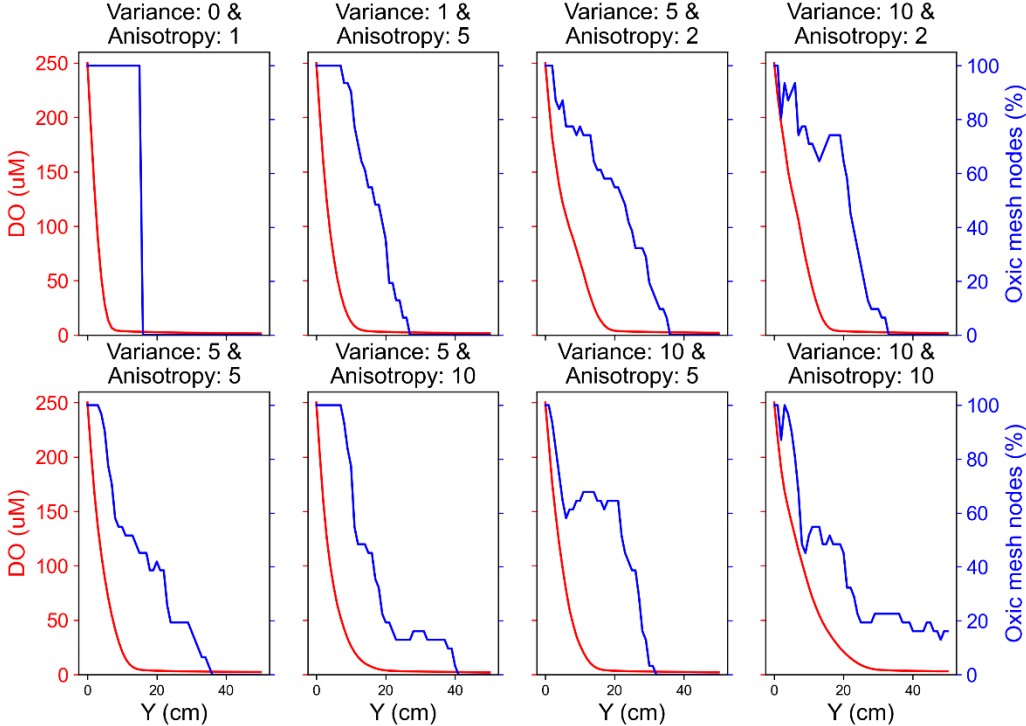

**Figure 3: Comparison of flux averaged DO concentration and % of oxic mesh nodes (i.e., cells with DO concentration > 3 µM) along the flow direction in a medium flow regime.**

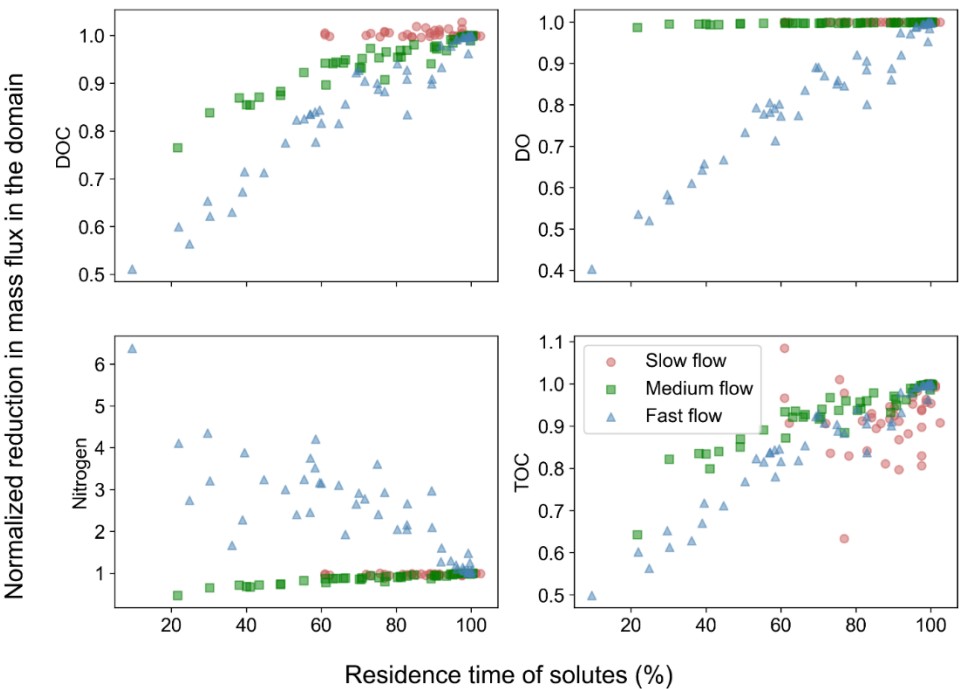

**Figure 4: Removal of chemical species in spatially heterogeneous domains in different flow regimes. Values show mass flux differences between inlet and outlet of the heterogeneous domains normalized by the flux differences for the homogeneous base case of each flow regime.**

The above results could be summarized by use of $\log_{10}Da$. The distribution of $\log_{10}Da$ is shown in Fig. S6. The same value of $\log_{10}Da$ is associated with different combinations of reaction and flow regimes. The aerobic reactions in the slow flow and the medium flow regimes were characterized with high values of $\log_{10}Da$ (>0.5), while the anaerobic reactions in the slow and medium flow regimes were characterized by mid-range values of $\log_{10}Da$ (0 - 0.5) along with the aerobic reactions in the fast flow regime. The anaerobic reactions in the medium flow regime were characterized by low values (-1 – 0) of $\log_{10}Da$. Lastly, the anaerobic reactions in the fast flow regime were characterized by extremely low values of $\log_{10}Da$ (<-1).

### 3.4 Distribution of microbial biomass in heterogeneous scenarios

As already shown in Sect. 3.1, the active immobile fraction of the biomass has a larger presence in the domain compared to the mobile fraction, thereby making a larger contribution to nutrient cycling. The median value of the mobile biomass in the domain varied from 98 μM C (in the fast flow regime) to 320 μM C (in the slow flow regime), out of which, active mobile biomass varied from 8 μM C (in the medium flow regime) to 15 μM C (in the fast flow regime). Immobile biomass in comparison was in the order of 300 μM in all flow regimes, out of which active immobile biomass varied from 29 μM (in the slow flow regime) to 232 μM (in the fast flow regime). Therefore, next we focus on the impact of spatial heterogeneity on the distribution of this important fraction of the biomass. Aerobic immobile degraders were found to be active and most abundant near the inlet of the domain, and along the preferential flow paths in the

downgradient zone of the domain (Fig. S4). Ammonia oxidizers were active at the interfaces between high
flow and low flow regions of the upstream parts of the system, co-existing with high concentration of active
aerobic degraders. Ammonia oxidizers were also active further downgradient in the system along the
preferential flow paths. This may be due to the presence of DO at reduced concentrations in the
downgradient region of the domain. DO at these concentrations and low DO/Ammonium ratios can be
preferentially taken up by ammonia oxidizers compared to aerobic degraders (Gu et al., 2006). The
maximum concentration of active immobile ammonium oxidizers was more than an order of magnitude
lower than that of the active immobile aerobic degraders. Nitrate reducers were present in lower
permeability zones in the heterogeneous domains, but close to the preferential flow paths, in response to
the continuous supply of nutrients from the groundwater flowing through the domain. They co-existed with
ammonia oxidizers but at higher concentrations. Active immobile nitrate reducers were much higher than
active mobile nitrate reducers in the fast and medium flow regimes, but comparable in magnitude in the
slow flow regime.

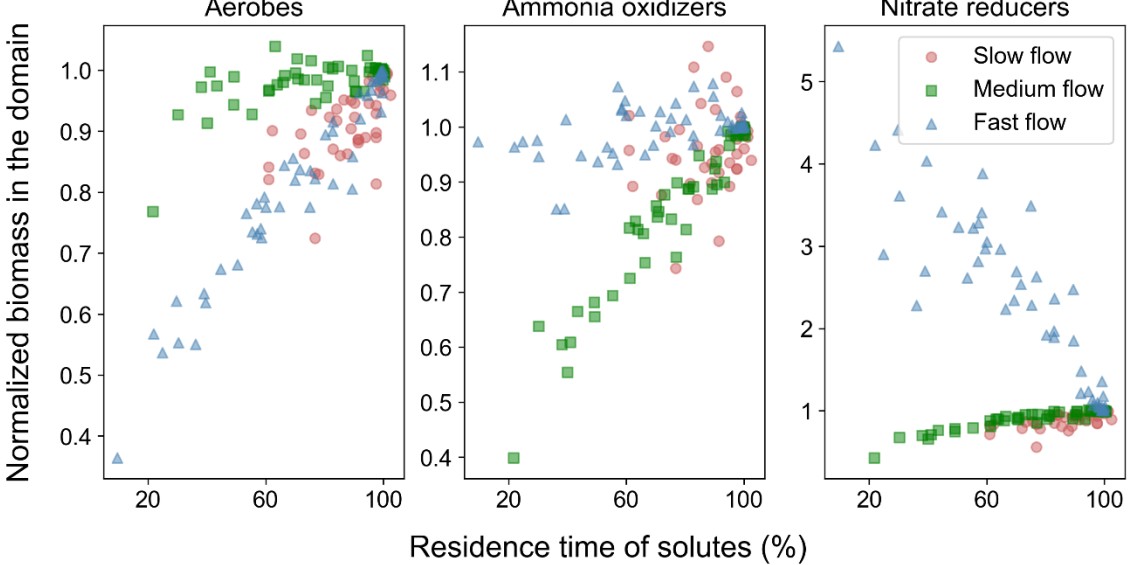

**Figure 5: Biomass concentration of active immobile fraction of different species in spatially heterogeneous domains in different flow regimes. Shown volume averaged values for the heterogeneous domains were normalized by values of the homogeneous base case of each flow regime.**

The corresponding 1-D distribution of microbial species in heterogeneous domains was observed to vary
from the base case given the same average water flux (Fig. S2). All the microbial species were prevalent
along a larger section in the heterogeneous domains. Aerobic and anaerobic microorganisms also appear to
co-exist in heterogeneous domains (solid lines), in contrast to their sequential occurrence in the base case
(dashed lines).

The changing distribution pattern of the microbial species impacts the total active immobile biomass
concentration in the domain, which diverges from the base case as heterogeneity increases (Fig. 5). The

biomass of active immobile aerobic degraders decreased with increasing heterogeneity regardless of the flow regime, with lowest values reaching only 40% of the base case biomass. The biomass of immobile active ammonia oxidizers and nitrate reducers also decreased with increasing heterogeneity in slow (~75% and ~90% of base case, respectively) and medium flow regimes (30% and 85% respectively). However, the impact on the biomass of immobile active nitrate reducers was the reverse in fast flow regime (increase to 5 times the concentration in the base case). Lastly, there was no impact of spatial heterogeneity in the biomass of immobile active ammonia oxidizers in the fast flow regime.

Overall, active immobile biomass decreased with increase in spatial heterogeneity in all the flow regimes, while active mobile biomass increased marginally (Fig. S9). Inactive immobile biomass reduced with spatial heterogeneity in slow and medium flow regimes, while it increased in the fast flow regime. Lastly, inactive mobile biomass increased with heterogeneity in all flow regimes.

**3.5 Predicting impact of spatial heterogeneity on redox regimes.**

While conducting the multivariate statistical analysis of change in mass removal of reactive species, we made use of AIC to evaluate governing factors influencing mass removal in a spatially heterogeneous domain. The analysis indicated that AIC was 994 when considering only breakthrough time and chemical species. AIC reduced to -211 when the chemical species, the flow regime, variance in permeability field and the anisotropy of the domain were included as random factors. Please refer to Table S1 for further details. Thus, we concluded that nutrient dynamics are influenced by spatial heterogeneity. Categorizing the systems using $\log_{10}Da$, we proposed a linear expression to predict the impact of spatial heterogeneity on nutrient removal. The regression parameters informing this expression are given in Table 5. The results indicated that we may underestimate nutrient removal by 6 times or overestimate it by twice the amount (Fig. 6).

**Table** 5**: Regression parameters for predicting removal of chemical reactive species based on the reaction regime indicated by $\log_{10}Da$.**

| Category of flow and reaction regime | Regression parameters | | |
|---|---|---|---|
| | **Slope** | **Intercept** | **RMSE** |
| $\log_{10}Da < -1$ | -365.0 | 497.3 | 57 |
| $-1 < \log_{10}Da < 0$ | 53.94 | 46.68 | 3.7 |
| $0 < \log_{10}Da < 0.5$ | 12.16 | 87.71 | 4.7 |
| $\log_{10}Da > 0.5$ | 0 | 100 | - |

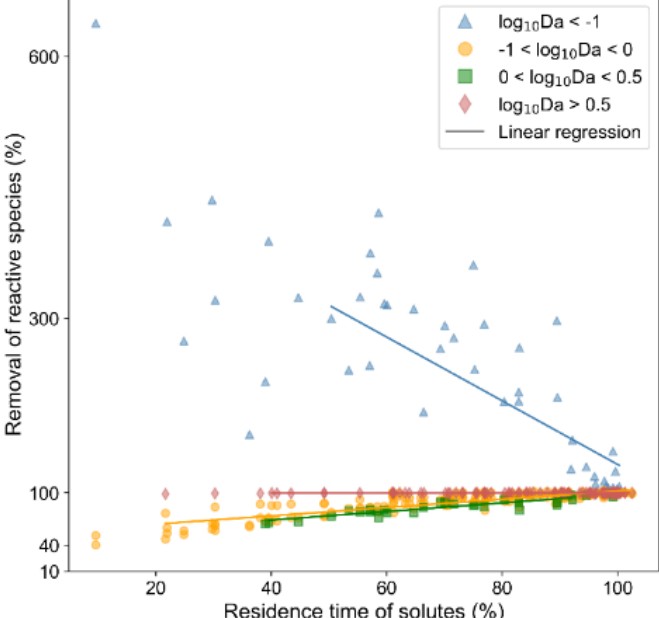

**Figure 6: Regression analysis: Predicting impact of spatial heterogeneity on chemical species removal in different reaction regimes indicated by $\log_{10}Da$. Value on Y-axis indicate the removal of chemical species in heterogeneous domains normalized by that in the corresponding base case. Spatial heterogeneity is plotted on the X-axis, indicated by the breakthrough time in the heterogeneous domain normalized by that in the base case (homogeneous domain). A value of 100% on the Y-axis indicates that the removal of the chemical species is the same as that in the corresponding base case (homogeneous domain). A value of 50% indicates that the removal of the chemical species reduced by half in the corresponding heterogeneous domain. A value of 600 indicates that the removal of the chemical species in the heterogeneous domain was 6 times that in the homogeneous domain.**

## 4 Discussion

In this study we synthesized available process knowledge and observations from our subject site on geomicrobial activity in the deep subsurface, both terrestrial and marine, into a set of in silico scenarios on the fate of biogeochemically reactive compounds in heterogeneous subsurface settings. This approach allowed us to generate a wide range of spatially heterogeneous domains (with variance of the log normal distribution of conductivity varying from 0.1 to 10, and anisotropy varying from 2 to 10), which is not possible experimentally. Therefore, we utilized geostatistical methods using variance in conductivity field and anisotropy to simulate heterogeneous subsurface scenarios. Variance reflects naturally occurring variation in the conductivity field. In case of a high variance, it represents scenarios where lenses of a different medium are present in another medium (such as clay lenses in a sandy aquifer). Anisotropy provides an additional control to enforce channelized flow fields in the domain or layering processes in general, common in both alluvial sediments and fractured bedrock. Thus, we considered 12 scenarios for representing these heterogeneous flow fields, covering most physically (variance) and geometrically (anisotropy) plausible scenarios. At the same time, linking extent of spatial heterogeneity with breakthrough

time allowed us to discuss the impact of spatial heterogeneity on removal of chemical species independently of how we generated the spatial heterogeneity.

The reaction network was formulated using literature knowledge and geomicrobial activity identified at the subject site. At the same time, it captures varying microbial respiration and growth regimes, from aerobic autotrophy to aerobic heterotrophy and anaerobic heterotrophy. The activity of geomicrobial reactive

systems is dependent on a variety of factors, such as nutrient availability, access to energy gradients, pH, pore size, hydraulic conductivity, particle size distribution (Smith et al., 2018). The limited information on microbial activity applicable to oligotrophic conditions in the subsurface does challenge the parameterization of the reaction network, which is a priori a potential major source of uncertainty for the obtained model results. Given this limitation, we calibrated the parameters of the reaction network to ensure

that it covers a sufficiently large range of Da values and that it does not violate the established redox hierarchy in any of the flow regimes considered (see Appendix A and the base case results). Additionally, we consistently used our parameter set in all scenarios and used results of the homogeneous base cases as internal reference to which we compared results of the individual heterogeneous scenarios as we aimed to study the impact of spatial heterogeneity on microbial activity and subsurficial nutrient dynamics.

Lastly, consideration of varying flow regimes in combination with the reaction network provides a view on both reaction dominant systems and flow dominant systems, indicated by the use of Da. This compensates for our approach wherein we do not explore additional scenarios varying concentrations of chemical species and their influence on microbial growth and distribution. By treating the analysis of results in terms of Da, we condense the discussion to effective rates of microbial activity given presence of spatial heterogeneity

of hydraulic conductivity. Thus, we are confident that the presented findings are not limited to the particular parameter set used in this study but that they are applicable widely.

### 4.1 Sampling and analysis: Biomass and reactive species

Microbial abundance can be derived from carbon content in the biomass using available conversion factors varying from 5 - 39 femtogram (fg) C/cell (Fukuda et al., 1998;Vrede et al., 2002). This resulted in median

values of total mobile biomass in the domain of $10^9$ to $10^{11}$ cells $L^{-1}$. Opitz et al. (2014) measured the total bacterial biomass in groundwater of the subject site to vary from $10^6$ to $10^8$ gene copies $L^{-1}$ (depending on location, tapped aquifer and season of measurement), which is lower than the simulated mobile values. However, the simulated values of mobile biomass are in the range derived in both lab scale and field scale studies (Holm et al., 1992;Griebler and Lueders, 2009;Grösbacher et al., 2018). Also, the mobile biomass

concentration is in the range of particulate organic carbon concentration observed to be exported in the seepage at the subject site (Lehmann et al., 2021). The relatively high biomass values obtained in the simulations are attributed to the relatively high inflow concentrations as well as to the relatively high

microbial reactivity we considered in the simulations to allow them to cover also high Da ranges. We note that while the total biomass may not be matching the observations at the subject site, the relative composition of the microbial species fractions (that is, immobile, mobile, active and inactive) follow established findings. For example, immobile microbial biomass indeed forms the majority biomass in the subsurface (as well as in our study), with its ratio with mobile biomass changing based on nutrient and other environmental conditions (Griebler et al., 2002; Grösbacher et al., 2018). It is proposed that the ratio of immobile and mobile biomass in (Griebler et al., 2002; Grösbacher et al., 2018) varies per nutrient availability, with higher ratios observed in oligotrophic conditions and lower ratios in nutrient rich conditions. We extend this further in our study, by observing that the ratio depends on the Damköhler number, with higher ratios in in low Da systems, and lower ratios in high Da (reaction dominant) systems. It is further estimated that 60%-80% of microbial biomass in soil may be inactive (Lennon and Jones, 2012). In our study, we observe these ranges in the slow flow and medium flow regimes, but not in the fast flow regime. With newer technologies equipped to better characterise activity of microbes in environmental samples (Couradeau et al. 2019), we expect that it will be easier to draw the comparison in the future.

It is also important to note that the estimated abundance at the subject site varies with both sampling location and season. And as mentioned above, we also observe that microbial biomass may be in different states of activity (active or inactive) or location (immobile or mobile) depending on the flow regime and the structure of spatial heterogeneity in the system. This brings into focus that next to spatial heterogeneities addressed in this study also temporal variations of environmental conditions can have a significant impact on microbial abundance (Eckert et al., 2015). This study provides preliminary insights into how varying water velocities/flow regimes may impact relative contribution of microbial species between inactive, active, mobile and immobile fractions in spatially heterogeneous domains. The system may respond similarly to temporal fluctuations in groundwater velocities resulting from seasonal cycles as well. While this is not part of the current study, the presented conceptual approach and assessment scheme may be applied in future studies focussing on such transient effects.

Commonly used groundwater sampling techniques do not resolve the heterogeneous distribution of chemical and microbial species along the length and cross-section of a well screen, though specialised probes exist to characterise small-scale chemical variability in the subsurface (Ronen et al., 1987). The obtained samples may thus present a skewed/biased observation of the biogeochemical dynamics in the subsurface. For example, the gradual reduction of flux averaged DO concentration from near saturation values at the inlet to below detection limit implies the continuous presence of an aerobic zone until DO is fully depleted. However, when the number of oxic and anoxic mesh nodes were calculated at each cross-section, it was evident that several oxic and anoxic regions can coexist in an apparently oxic zone (Fig. 4). This results in unexpected observations wherein aerobes and anaerobes appear to be active in similar

conditions, while in fact, their zones of activity are spatially separated, and anaerobes are active in smaller zones that specifically provide hospitable conditions to their activity. Spatial heterogeneity allows for this apparent co-occurrence of several microbial species by providing appropriate niches. For instance, the

biomass of immobile active nitrate reducers increased with spatial heterogeneity in the fast flow regime due to the introduction of sub-oxic pockets with anaerobic activity in low flow zones within a predominantly oxic zone with aerobic activity (Fig. S4). Seemingly overlapping conditions have also been observed by field scale studies (Alewell et al., 2006;Waldron et al., 2009;Schwab et al., 2017;Lohmann et al., 2020), although the diversity of microbial communities varied in both space and time. Alewell et al., (2006),

Schwab et al. (2017) and Lohmann et al. (2020) also noted that small scale heterogeneities did not allow for the sequential redox hierarchy (as defined by energy yields of redox half reactions) to be applicable at the meter scale. We establish that the persistence of microbial species in the domain is governed by the presence of the appropriate carbon source and electron acceptor, despite apparent co-existing microbial species that may be identified by groundwater sampling techniques that do not resolve sub-sampling scale

heterogeneities. Therefore, while mobility of microbial species using water as the medium may temporarily affect the composition of microbial communities, it is unlikely that mobile microbial species persist in high numbers at a location in absence of sustained sources of nutrient and energy. This is further evident from the impact of spatial heterogeneity on microbial biomass distribution whereby active microbial biomass is only found to be persistent in high numbers in zones where reactive species are easily accessible. In

addition, Kim et al. (2009) and Kim et al. (2019) also suggested that groundwater redox chemistry and distribution of carbon pools are linked with geological controls such as hydraulic conductivity. The requirement of vertically discretized sampling has already been recognized (Ronen et al., 1987; Smith et al., 2018) and addressed by various sampling methodologies such as low flow sampling techniques, passive samplers, point and discrete interval samplers (Ronen et al., 1987;Smith et al., 1991;Powell and Puls,

1993;Báez-Cazull et al., 2007;Anneser et al., 2008) even though heterogeneities at scales lower than that resolved by the sampling scheme will remain unobserved. Our results support the usefulness of such spatially resolved sampling techniques for analysis of microbial activity in the groundwater. On the other hand, composite sampling from macro-scale matrix samples is useful to estimate the microbial activity in the sampled matrix core. This enables a more accurate estimate of microbial activity aggregated over the

matrix core.

The impact of heterogeneity on microbial biomass distribution has strong implications for evaluating sampling techniques and data obtained from groundwater samples. Immobile microbes account for more microbial activity compared to mobile microbes. However, groundwater samples represent mobile microbial biomass, termed as planktonic biomass (Smith et al., 2018). Estimates of microbial respiration

are thereafter made based on the abundance of mobile microbes in the obtained groundwater samples. The

results of this study suggest that the immobile microbes are, in fact, the major contributors to microbial respiration in the subsurface, which was also experimentally established by (Alfreider et al., 1997;Griebler et al., 2002;Grösbacher et al., 2018) by providing the link between heterogeneous structures in the domain, corresponding nutrient availability and microbial biomass growth. Our results suggest that the relative composition of species in the mobile and immobile subcommunities is however similar (in part due to the same detachment/attachment properties assigned to the microbial species in this study), such that the assessment of microbial diversity based on the mobile fraction, only, would still be representative. However, this is not necessarily the case for nutrient cycling (see below).

Since spatial heterogeneity impacts microbial biomass distribution and microbial activity, it is not surprising that spatial heterogeneity also impacts carbon and nitrogen removal. Sanz-Prat et al. (2015, 2016) already established that travel time models are valid for use as reactive transport models in steady-state advective-transport conditions with other studies already discussing the same in surface waters and hyporheic zones (Liao and Cirpka, 2011;Painter, 2018). Painter (2018) also considers the application of travel time distribution as a representation of heterogeneity to be specific to the processes or reactive species being considered. We further this understanding by exploring a wider range of flow regimes, from locally mixed regimes to dominantly advective flow regimes and a complex process network exploring a variety of reactive species across both aerobic and anaerobic microbial processes. The impact on removal of carbon resulting from heterogeneity is consistent for all flow regimes, with carbon removal decreasing in heterogeneous domains. For nitrogen removal the same trend is observed for the slow and medium flow regimes. In contrast, nitrogen removal in the fast flow regime increases with spatial heterogeneity, as spatially heterogeneous domains provide the opportunity for anaerobic activity sub-zones to sustain in predominantly oxic systems with aerobic regimes. As for the fast flow regime oxic conditions prevailed until the vicinity of the outlet of the simulated domain, nitrogen removal is mainly restricted to such sub-oxic sub-zones and heterogeneity leads to an increased number of such subzones. It must be noted though that the concentration of nitrate decreases when and where the concentration of DO is below 15 µM (Fig. S1) (De Brabandere et al., 2014;Kalvelage et al., 2013;Seitzinger et al., 2006). The reduced concentration of nitrate is attributable to the activity of nitrate reducers. It is assumed that for sufficiently long domains with sub-oxic conditions dominating the downstream parts nitrogen removal would also exhibit a decreasing trend with heterogeneity, even for the fast flow regime. Therefore, travel time information is useful for estimating both carbon and nitrogen removal and identifying dominant microbial redox processes despite sub-scale heterogeneities allowing for co-existence of several microbial species. Since immobile active microbial biomass was the major contributor to carbon and nitrogen removal, the reduction in removal of reactive species can be traced to reduced presence of immobile active microbial biomass in heterogeneous domains. In contrast, the contribution of the mobile active biomass in heterogeneous

660    domains remains largely the same as that in homogeneous domains. This indicates that the mobile microbial abundance detected in groundwater samples must be used with care as a proxy for effective microbial activity and nutrient cycling (also confirmed by Alfreider et al. (1997), Murphy et al., (1997), Griebler et al. (2002) and Grösbacher et al. (2018) as mentioned earlier).

**4.2 Indicators to evaluate impact of spatial heterogeneity on biomass and redox regimes**

In this study, we explored three different flow regimes, representing Peclet (Pe) numbers varying over an order of magnitude. The Damköhler number of the varying components of the system (derived from the observed mass removals and breakthrough times) varies over 4 orders of magnitude (Fig. S6 (a)) in the considered scenarios.

There is substantial overlap in Da across all the flow regimes; a given reactive species has different Da in 670    the different heterogeneous domains in each flow regime. However, spatial heterogeneity impacts the removal of each reactive species in the flow regimes differently. This is further evidenced in the significant improvement of the model AIC (Table S1) when the reactive species is included as a fixed effect in conjunction with the flow regime. While this approach helps us to generate a predictive understanding of system behaviour, it is specific to reactive species and flow regimes concerned. For a scalable approach to 675    modelling and predicting nutrient cycles at larger scales, it is therefore useful to consider proxy indicators that may assist in generalizing this expression.

Noting that the impact of spatial heterogeneity on removal of nitrogen in the medium flow regime, and of DO and TOC in the fast flow regime is the same given the same reduction in breakthrough time (Fig. S7), we consider the impact of spatial heterogeneity in context of Da or $\log_{10}$Da, thus providing an opportunity 680    to disentangle reactive species and flow regimes in terms of non-dimensional numbers (Fig. 5). The impact of spatial heterogeneity on nutrient cycling varies with the value of $\log_{10}$Da (Fig. 5). For values higher than 0.5, the impact is negligible. For $\log_{10}$Da < -1, spatial heterogeneity results in an increased removal of nutrients from the domain. This is in part due to negligible removal of the corresponding nutrients in the base case (specifically, nitrogen in the fast flow regime, refer to Sect. 3.3 and 4.1). Even a marginal increase 685    in the relevant microbial activity results in a remarkably high impact on the removal of the corresponding nutrient when compared to the base case. As discussed above, for the fast flow regime oxic conditions with aerobic activity are found along the entire homogeneous domain. Since most nitrogen removal processes are suppressed by elevated DO concentrations, the formation of a sub-oxic zone exhibiting anaerobic activity in low flow regions of the heterogeneous domains is the only chance of these nitrogen removal 690    processes to take place in the fast flow scenarios. The observed increase in mass removal with heterogeneity is thus to some extent an artefact that may not represent a general trend. While heterogeneity does have an

impact on nitrogen removal in the fast flow regime, even after increased removal, the $\log_{10}Da$ value remains below -1, indicating low absolute activity/removal.

For regimes where $\log_{10}Da > 0.5$, spatial heterogeneity has limited impact on the ability of the system to remove reactive species. But, removal of reactive species decreases with reduction in breakthrough time for $-1 < \log_{10}Da < 0.5$. To explore the cause of this, we compared the trend of removal of reactive species in first order rates (Fig. S8) and zero order rates with reduced residence times with the simulation results for varying values of $\log_{10}Da$. The mean of $\log_{10}Da$ in this regime was -0.3 with a standard deviation of 0.3. So, we approximated the analytical solution for varying values of $\log_{10}Da$ (Fig. S8). With increasing $\log_{10}Da$ (between 0 and 0.5), the root mean squared error (RMSE) between the analytical solution for a first order reaction and the simulation results decreases. Additionally, the data points lie in between the solutions for first order and zero order kinetics, as it would be the case for Monod kinetics in case of reduced residence times. Consequently, the impact of spatial heterogeneity on regimes with $0 < \log_{10}Da < 0.5$ may be described on the bases of reducing residence time alone and the results do not allow to determine if additional heterogeneity effects on removal take place. For regimes where $-1 < \log_{10}Da < 0$, first order kinetics may be substituted with zero order kinetics. Additionally, the impact on mass removal of reactive species in this domain is lower than estimated from the analytical solution. Therefore, while mass removal of reactive species reduces with reducing breakthrough times, it does not follow Monod kinetics which implies that heterogeneity has a different impact on removal than changing only the residence time. In fact, the impact of spatial heterogeneity on mass removal is lower than that predicted by reducing residence time alone. For a quantitative assessment we proposed linear regression metrics to estimate mass removal resulting from reducing residence times. At the same time, we observed that the dramatic increase in mass removal for regimes $\log_{10}Da < -1$ is not attributable to a shorter residence time, but due to heterogeneous conditions providing niches to the relevant microbial species to become active. Therefore, we conclude that spatial heterogeneity may result in changed nutrient dynamics. The regression model links the impact of heterogeneity to variables which can be estimated in field studies. Furthermore, this helps to categorize reaction regimes to consider if spatial heterogeneity is of significance. For high $\log_{10}Da$ values, spatial heterogeneity is not of significance. For extremely low $\log_{10}Da$ values, spatial heterogeneity resulted for the used model domains in a high impact on removal rates with respect to the homogeneous base cases. However, this might be an artifact of the used domain size and the absolute removal values are still low. Thus, heterogeneity effects may be neglected for these low $\log_{10}Da$ values. In turn, spatial heterogeneity is significant for medium range $\log_{10}Da$ values ($-1 < \log_{10}Da < 0.5$). For these values, the highest heterogeneity induced reductions in mass removal were observed and can be well described by the linear regression model.

We expect advection dominated systems to be impacted by spatial heterogeneity because spatial heterogeneity had a higher impact on the transport profiles in these systems. These are typically systems that are shallow, less compacted (in case of alluvial sediments), or fractured rock systems. Furthermore, the shallow subsurface also receives bioavailable and reactive organic matter with the incoming water which enables a relatively high microbial activity. In contrast, in the deep subsurface microbial activity is lower

and rather relies more often on inputs from the matrix material, which is ubiquitous and does not rely on transport for access to nutrients or energy gradients. It must be noted that this generic description of dominant processes serves to give examples for reactive systems for the purpose of our discussion and may vary from site-to-site depending on specific site characteristics. We expect additional studies exploring the impact of varying concentrations of chemical species, parameters relevant to these ecosystems or subject

sites to add to the evidence generated by our study that the impact of spatial heterogeneity on subsurficial reactive systems may be predicted using field estimated indicators such as breakthrough time, Pe and Da.

**5 Summary and Conclusions**

In this study, we investigated the impact of spatial heterogeneity on biomass persistence, distribution, and nutrient cycling at the sub-meter scale in the subsurface. When considering spatial heterogeneity, a

740 combination of variance and anisotropy of the hydraulic conductivity was considered when evaluating the transport regime, which may be further interpreted as a reduction of solute residence time in the domain. The flow regime was found to play an influential role in the average behaviour of the domain. Not only does the total microbial biomass vary with the flow regime, but the contribution of different fractions of microbial biomass (between active or inactive, mobile or immobile) is also different based on the flow

regime. Spatial heterogeneity also impacts the different fractions of microbial biomass differently. This has a cumulative impact on nutrient cycling in the subsurface. The activity of the microbial species in the domain is governed by the spatial heterogeneity as it influences the distribution of nutrients and energy sources. We found that several microbial species that are conventionally accepted to occupy mutually exclusive niches may co-exist in the subsurface in close vicinity. This further demonstrates that the

occurrence of oxic systems does not preclude the existence of anaerobic species in the same zone as heterogeneity leads to the formation of sub-oxic regions with anaerobic activity within an oxic zone exhibiting predominantly aerobic activity. Since modelers and experimentalists do not conventionally resolve these small-scale heterogeneities the accuracy of the prediction of biogeochemical cycles at the larger scale suffers.

Depending on the reaction and flow regime of the domain, the impact of spatial heterogeneity on mass removal of reactive species can be quantified as a linear function of the breakthrough time. We propose the

use of the Damköhler number to identify the appropriate parameters of this function. Simulations that neglect or aggregate microbially mediated dynamics in spatially heterogeneous media may overestimate reactive species removal by as much as 2 times. This factor can be predicted using readily observable data that informs Damköhler numbers and residence times using a linear function of residence time. We propose using this scaling factor to account for heterogeneity in regional scale simulations for accurate prediction of microbial mediated reactive species dynamics in groundwater.

## Appendix A: Biochemical reaction network

### A.1 Reactive Species

1. Chemical compounds:
    a. Dissolved Organic Carbon (DOC)
    b. Particulate Organic Carbon (POC)
    c. Oxygen (O2)
    d. Nitrate (NO3)
    e. Sulphate (SO4)
    f. Ammonium (NH4)
2. Microbial species:
    a. Aerobic DOC degraders (BO2)
    b. Nitrate reducers (BNO3)
    c. Sulphate reducers (BSO4)
    d. Ammonia oxidizers (BNH4)

For each microbial species, we considered different subpopulations: active bacteria able to grow and to perform biogeochemical reactions, inactive bacteria, immobile bacteria attached to the solid matrix, mobile bacteria moving with the flowing water. In combination this leads to four subpopulations for each microbial species X: active immobile ($X_{a,s}$), active mobile ($X_{a,w}$), inactive immobile ($X_{i,s}$) and inactive mobile ($X_{i,w}$).

### A.2. Biogeochemical Reactions

| | | |
|---|---|---|
| Aerobic respiration: | $CH_2O + O_2 \rightarrow HCO_3^- + H^+$ | (A1) |
| Nitrate reduction: | $CH_2O + 0.8NO_3^- + 0.8H^+ \rightarrow HCO_3^- + 0.4N_2 + 0.4H_2O + H^+$ | (A2) |
| Sulphate reduction: | $CH_2O + 0.5SO_4^{2-} + H^+ \rightarrow HCO_3^- + 0.5HS^- + 1.5H^+$ | (A3) |
| Ammonia oxidation: | $0.5NH_4^+ + O_2 \rightarrow 0.5NO_3^- + 0.5H_2O + H^+$ | (A4) |
| Hydrolysis of POC: | $C_{10}H_7O_2N + 8H_2O + H^+ \rightarrow 10CH_2O + NH_4^+$ | (A5) |

### A.3. Rate expressions:

### A.3.1 Microbial respiration

We used modified Monod-type expressions for microbially driven reactions:

1. Aerobic respiration:

$$r = \frac{kmax1 \times \left(\frac{DOC}{ksodoc+DOC}\right) \times \left(\frac{O2}{ksox+O2}\right)}{e^{\frac{doxmin-kmax1 \times \left(\frac{DOC}{ksodoc+DOC}\right) \times \left(\frac{O2}{ksox+O2}\right)}{st \times doxmin}}+1} \left(BO2_{a,s} + BO2_{a,w}\right) \tag{A6}$$

2. Nitrate reduction:

$$r = \frac{kmax2 \times \left(\frac{DOC}{ksodoc+DOC}\right) \times \left(\frac{NO3}{ksno3+NO3}\right) \times \left(\frac{kindox}{kindox+O2}\right)}{e^{\frac{no3min-kmax2 \times \left(\frac{DOC}{ksodoc+DOC}\right) \times \left(\frac{NO3}{ksno3+NO3}\right) \times \left(\frac{kindox}{kindox+O2}\right)}{st \times no3min}}+1} \left(BNO3_{a,s} + BNO3_{a,w}\right) \tag{A7}$$

3. Sulphate reduction:

$$r = \frac{kmax3 \times \left(\frac{DOC}{ksodoc+DOC}\right) \times \left(\frac{SO4}{ksso4+SO4}\right) \times \left(\frac{kindox}{kindox+O2}\right) \times \left(\frac{kinno3}{kinno3+NO3}\right)}{e^{\frac{so4min-kmax3 \times \left(\frac{DOC}{ksodoc+DOC}\right) \times \left(\frac{SO4}{ksso4+SO4}\right) \times \left(\frac{kindox}{kindox+O2}\right) \times \left(\frac{kinno3}{kinno3+NO3}\right)}{st \times so4min}}+1} \left(BSO4_{a,s} + BSO4_{a,w}\right) \tag{A8}$$

4. Ammonia oxidation:

$$r = \frac{kmax4 \times \left(\frac{O2}{ksox+O2}\right) \times \left(\frac{NH4}{ksamm+NH4}\right)}{e^{\frac{nh4min-kmax4 \times \left(\left(\frac{O2}{ksox+O2}\right)\right) \times \left(\frac{NH4}{ksamm+NH4}\right)}{st \times nh4min}}+1} \left(BNH4_{a,s} + BNH4_{a,w}\right) \tag{A9}$$

### A.3.2 Microbial growth

Growth processes (i.e. formation of biomass carbon) are linked to rates of microbially driven reactions using a constant yield factor with an additional dependency on the concentration of ammonium, and its availability for uptake. For the latter, we considered a fixed ratio between carbon (DOC) and nitrogen (NH4) uptake.

1. Dependency on ammonium:

$$NH4limit = \frac{1}{e^{\frac{amming-NH4}{st \times amming}}+1} \tag{A10}$$

2. Active aerobic DOC degraders:

$$r = NH4limit \times \frac{kmax1 \times \left(\frac{DOC}{ksodoc+DOC}\right) \times \left(\frac{O2}{ksox+O2}\right)}{e^{\frac{O2min-kmax1 \times \left(\frac{DOC}{ksodoc+DOC}\right) \times \left(\frac{O2}{ksox+O2}\right)}{st \times O2min}}+1} Y_o \times BO2_{a,x} \tag{A11}$$

with a = active biomass, x=s for attached and x=w for mobile bacteria

3. Active nitrate reducers:

$$r = NH4limit \times \frac{kmax2 \times \left(\frac{DOC}{ksodoc+DOC}\right) \times \left(\frac{NO3}{ksno3+NO3}\right) \times \left(\frac{kindox}{kindox+O2}\right)}{e^{\frac{no3min-kmax2 \times \left(\frac{DOC}{ksodoc+DOC}\right) \times \left(\frac{NO3}{ksno3+NO3}\right) \times \left(\frac{kindox}{kindox+O2}\right)}{st \times no3min}}+1} Yn \times BNO3_{a,x} \tag{A12}$$

4. Active sulphate reducers:

$$r = NH4limit \times \frac{kmax3 \times \left(\frac{DOC}{ksodoc+DOC}\right) \times \left(\frac{SO4}{ksso4+SO4}\right) \times \left(\frac{kindox}{kindox+O2}\right) \times \left(\frac{kinno3}{kinno3+NO3}\right)}{e^{\frac{so4min-kmax3 \times \left(\frac{DOC}{ksodoc+DOC}\right) \times \left(\frac{SO4}{ksso4+SO4}\right) \times \left(\frac{kindox}{kindox+O2}\right) \times \left(\frac{kinno3}{kinno3+NO3}\right)}{st \times so4min}}+1} \; Ys \; \times BSO4_{a,x}$$

(A13)

5. Active ammonia oxidizers:

$$r = NH4limit \times \frac{kmax4 \times \left(\frac{O2}{ksox+O2}\right) \times \left(\frac{NH4}{ksamm+NH4}\right)}{e^{\frac{nh4min-kmax4 \times \left(\left(\frac{O2}{ksox+O2}\right)\right) \times \left(\frac{NH4}{ksamm+NH4}\right)}{st \times nh4min}}+1} \; Ya \; \times BNH4_{a,x}$$

(A14)

### A.3.3 Processes governing the location of the microbes

1. Mobilization of immobilized bacteria (Bxx) into the fluid medium (i.e. the transfer of attached bacteria into mobile bacteria) are adapted from (Rittman and McCarty (2001)) assuming additionally that high total attached biomasses lead to higher detachment rates (adapted from Clément et al. (1997)):

$$r = kl \; \times \; (vq0 \times vpor0)^{0.58} \times Bxx \; +$$

$$\frac{kdet}{e^{\frac{Bfmax-Bo2_{a,s}-BO2_{i,s}-BNO3_{a,s}-BNO3_{i,s}-BSO4_{a,s}-BSO4_{i,s}-BNH4_{a,s}-BNH4_{i,s}}{st \times Bfmax}}+1} \times Bxx$$

(A15)

2. Immobilization or reattachment: Attachment rates of mobile bacteria Byy is also depending on the total concentration of attached biomass:

$$r = \; katt \times \left(1 - \frac{1}{e^{\frac{Bfmax-Bo2_{a,s}-BO2_{i,s}-BNO3_{a,s}-BNO3_{i,s}-BSO4_{a,s}-BSO4_{i,s}-BNH4_{a,s}-BNH4_{i,s}}{st \times Bfmax}}+1}\right) \times Byy$$

(A16)

### A.3.4 Processes governing the activity states of microbes:

1. Deactivation/Dormancy: Deactivation rates of active bacteria (i.e., conversion of active (mobile/attached) into inactive or inactive (mobile/attached) bacteria) at unfavourable substrate conditions are expressed following Stolpovsky et al. (2011).

$$r = kdeac \; \times Bxx \; \times (1 - \frac{1}{e^{\frac{Kxx}{st}}+1})$$

(A17)

with the term Kxx depending on the bacterial species Byy and its substrate source (see Table A1).

2. Reactivation: In analogy to the deactivation rates, reactivation rates are expressed as:

$$r = kreac \; \times Byy \times \frac{1}{e^{\frac{Kxx}{st}}+1}$$

(A18)

with the term Kxx depending on the bacterial species as described in Table A1.

3. Mortality: Mortality rates follow a first-order dependency on biomass concentration:

$$r = km \times fdorm \times Bxx \tag{A19}$$

For active bacteria fdorm = 1, for inactive bacteria fdorm = 0.1. Dead bacterial biomass is added to the POM pool.

**Table A1 Expressions controlling respiration, growth, dormancy, and reactivation of microbial species.**

| Notation | Descriptors | | | |
| --- | --- | --- | --- | --- |
| | **Aerobic degraders** | **Nitrate reducers** | **Sulphate reducers** | **Ammonia oxidizers** |
| Bxx | $BO2_{a,s}$ and $BO2_{a,w}$ | $BNO3_{a,s}$ and $BNO3_{a,w}$ | $BSO4_{a,s}$ and $BSO4_{a,w}$ | $BNH4_{a,s}$ and $BNH4_{a,w}$ |
| Byy | $BO2_{i,s}$ and $BO2_{i,w}$ | $BNO3_{i,s}$ and $BNO3_{i,w}$ | $BSO4_{a,s}$ and $BSO4_{i,w}$ | $BNH4_{a,s}$ and $BNH4_{i,w}$ |
| Kxx | $1 - kmax1$ $\times \left(\frac{DOC}{ksodoc1 + DOC}\right)$ $\times \left(\frac{O2}{ksox1 + O2}\right)$ $/O2min$ | $1 - kmax2$ $\times \left(\frac{DOC}{ksndoc + DOC}\right)$ $\times \left(\frac{kindox}{kindox + O2}\right)$ $\times \left(\frac{NO3}{ksno3 + NO3}\right)$ $/NO3min$ | $1 - kmax3$ $\times \left(\frac{DOC}{kssdoc + DOC}\right)$ $\times \left(\frac{SO4}{ksso4 + SO4}\right)$ $\times \left(\frac{kindox}{kindox + O2}\right)$ $\times \left(\frac{kinno3}{kinno3 + NO3}\right)$ $/SO4min$ | $1 - kmax4$ $\times \left(\frac{NH4}{ksamm + NH4}\right)$ $\times \left(\frac{O2}{ksox + O2}\right)$ $/NH4min$ |

### A.3.5 Miscellaneous processes:

1. Hydrolysis of POC is described by first order rate kinetics:

$$r = kpd \times POC \tag{A20}$$

2. Background autotrophic microbial growth dependent on the presence of ammonium:

$$r = NH4limit \times kmax5 \times \left(\frac{NH4}{ksamm+NH4}\right) \tag{A21}$$

### A.4. Parameters

**A.4.1 Biogeochemical reaction network parameters**

| S. No. | Description | Notation | Value | Units | Source |
| --- | --- | --- | --- | --- | --- |
| 1 | Rate constant for aerobic reduction of DOC | kmax1 | 1 | d$^{-1}$ | calibrated |

| S. No. | Description | Notation | Value | Units | Source |
|---|---|---|---|---|---|
| 2 | Minimum biomass normalized rate value for aerobic respiration to be favourable | o2min | 0.06 | $d^{-1}$ | calibrated |
| 3 | Yield coefficient for growth of aerobic degraders of DOC | Yo | 0.25 | - | calibrated, based on Thullner et al., 2005 |
| 4 | Half-velocity DOC concentration | ksodoc | 1,000 | µM C | calibrated, based on concentrations observed in the field |
| 5 | Half velocity oxygen concentration | ksox | 2- | µM | Thullner et al. (2005), Wang and van Cappellen (1996) |
| 6 | Rate constant for nitrate reduction | kmax2 | 0.9 | $d^{-1}$ | calibrated, based on Schäfer et al.(1998b) |
| 7 | Minimum biomass normalized rate value for respiration to be favourable | no3min | 0.1 | $d^{-1}$ | calibrated |
| 8 | Yield coefficient for growth of nitrate reducers | Yn | 0.17 | - | calibrated, based on Clément et al. (1997) and Thullner et al. (2005) |
| 9 | Half-velocity DOC concentration | ksndoc | 1,000 | µM C | calibrated, based on concentrations observed in the field |
| 10 | Half velocity nitrate concentration | ksno3 | 100 | µM | calibrated, based on Clément et al. (1997) and André et al. (2011) |
| 11 | Inhibition constant for presence of oxygen | kindox | 1 | µM | calibrated, based on detection limits of sensors defining anaerobic conditions |
| 12 | Rate constant for sulphate reduction | kmax3 | 0.03 | $d^{-1}$ | calibrated |

| S. No. | Description | Notation | Value | Units | Source |
|---|---|---|---|---|---|
| 13 | Minimum biomass normalized rate value for respiration to be favourable | so4min | 0.0039 | d$^{-1}$ | calibrated |
| 14 | Yield coefficient for growth of sulphate reducers | Ys | 0.02 | - | calibrated, based on Thullner et al. (2005) |
| 15 | Half-velocity DOC concentration | kssdoc | 1,000 | µM C | calibrated, based on concentrations observed in the field |
| 16 | Half velocity sulphate concentration | ksso4 | 1,000 | µM | calibrated, based on Pallud and Van Cappellen (2006), Thullner et al. (2005) and Boudreau and Westrich (1984) |
| 17 | Inhibition constant for presence of oxygen | kindox | 1 | µM | calibrated, based on detection limits of sensors defining anaerobic conditions |
| 18 | Inhibition constant for presence of nitrate | kinno3 | 50 | µM | calibrated |
| 19 | Rate constant for ammonia oxidation | kmax4 | 0.1 | d$^{-1}$ | calibrated |
| 20 | Minimum biomass normalized rate value for respiration to be favourable | ammin | 0.004 | d$^{-1}$ | calibrated |
| 21 | Yield coefficient for growth of ammonia oxidizers | Ya | 0.0038 | | calibrated |
| 22 | Half-velocity Ammonia concentration | ksamm | 20 | µM | calibrated based on conditions observed in the field |
| 23 | Half velocity oxygen concentration | ksox | 20 | µM | De Brabandere et al. (2014), Kalvelage et al. (2013), Seitzinger et al. (2006) |

| S. No. | Description | Notation | Value | Units | Source |
|--------|-------------|----------|-------|-------|--------|
| 24 | Maximum/Carrying capacity at a node | Bfmax | 500 | µM C | calibrated, based on Fukuda et al. (1998), Vrede et al. (2002) and Grösbacher et al. (2018). |
| 25 | Mobilisation rate constant due to exceedance of carrying capacity | kdet | 1 | µM C $d^{-1}$ | calibrated, based on Clément et al. (1997) |
| 26 | Sigmoidal function slope parameter | st | 0.1 | - | Stolpovsky et al. (2011) |
| 27 | Minimum concentration of Ammonium for growth to remain favourable | amming | 10 | µM | calibrated based on review by Jin et al. (2013) |
| 28 | Immobilisation rate constant | katt | 0.3 | µM C $d^{-1}$ | calibrated, based on kdet |
| 29 | Deactivation/dormancy rate constant | kdeac | 1 | $d^{-1}$ | calibrated, based on Stolpovsky et al. (2016) |
| 30 | Reactivation rate constant | kreac | 0.3 | $d^{-1}$ | calibrated, based on Stolpovsky et al (2016). |
| 31 | Mortality rate constant | km | 0.01 | $d^{-1}$ | calibrated, based on Clément et al. (1997) |
| 32 | Hydrolysis constant | kpd | 0.03 | $d^{-1}$ | calibrated |
| 33 | Carbon Nitrogen ratio for hydrolysis of particulate organic matter | fcn | 10:1 | - | calibrated, based on Wang and van Cappellen (1996) |
| 34 | Desorption constant | kl | 0.00544 | - | Rittmann and McCarty (2001) |
| 35 | Rate constant for background activity | kmax5 | 0.00038 | $d^{-1}$ | calibrated |

**Table A2 Parameters used for the biogeochemical reaction network**

### A.4.2 Transport boundary conditions

| Property (units) | Value |
|------------------|-------|
| Porosity (-) | 0.2 |

| | |
|---|---|
| Density (matrix, kg m$^{-3}$) | 1,500 |
| Density (groundwater, kg m$^{-3}$) | 1,000 |

**Boundary conditions for chemical and microbial species at the inlet of the domain**

| | |
|---|---|
| POM (µM C) | 5 |
| DOC (µM C) | 800 |
| DO (µM) | 250 |
| Nitrate (µM) | 250 |
| Ammonium (µM) | 60 |
| Sulphate (µM) | 1,500 |
| Bx,w (µM C) | 2 |

**Table A3 Flow and transport parameterization and boundary conditions for all domains in all three flow regimes. The boundary condition for the reactive species was a Dirichlet boundary condition (fixed concentration) at the inlet of the domain.**

**Code Availability**

The source code of OGS5 is available in an online repository (Khurana et al., 2021). The input files for all simulated scenarios are available on the same repository. In addition, we also provide the Maple worksheet and resulting BRNS dynamically linked library that is required to run the simulations in OGS#BRNS at this link. We also provide the Python scripts used for processing the raw simulation results and for generating graphics in this publication at this link. The processed datasets are also available in this repository.

**Data Availability**

We provide the raw simulation results on this repository on Zenodo (Khurana et al., 2020) along with processed data files.

**Author Contribution**

Swamini Khurana conceptualized the reaction network, defined the scenarios to be simulated, executed the simulations, analysed the results and drafted the manuscript. Anke Hildebrandt, Falk Heße and Martin Thullner provided guidance and technical support. Martin Thullner defined and supervised the study. All co-authors contributed to the interpretation of the results and editing the manuscript.

**Competing interests**

The authors declare that they have no conflict of interest.

**Acknowledgements**

This study is part of the Collaborative Research Centre AquaDiva of the Friedrich Schiller University Jena,
funded by the Deutsche Forschungsgemeinschaft (DFG, German Research Foundation) – SFB 1076 – Project Number 218627073. The authors would like to thank Thomas Kalbacher for his valuable support in setting up the scenarios in OpenGeoSys, and Tino Rödiger for providing useful insight into the recharge dynamics at and geology of the subject site.

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
