# Peer review of "Predicting the impact of spatial heterogeneity on microbially mediated nutrient cycling in the subsurface"

_Biogeosciences, 2021_

## Author Comment (AC1)

**Predicting the impact of spatial heterogeneity on microbial redox dynamics and nutrient cycling in the subsurface**

Swamini Khurana[1], Falk Heße[2,3], Anke Hildebrandt[2,4,5], Martin Thullner[1]

[1]Department of Environmental Microbiology, Helmholtz Centre for Environmental Research – UFZ, Leipzig, 04318, Germany
[2]Department of Computational Hydrosystems, Helmholtz Centre for Environmental Research – UFZ, Leipzig, 04318, Germany
[3]Institute of Earth and Environmental Sciences, University Potsdam, Potsdam, Germany
[4]Institute of Geoscience, Friedrich-Schiller-University Jena, Jena, Germany
[5]German Centre for Integrative Biodiversity Research, Leipzig, Germany

**Correspondence**: Swamini Khurana (swamini.khurana@ufz.de)

**Response to RC1**

*We thank the reviewer for the constructive comments. We acknowledge that we have to provide an improved explanation of the aims of our study and of the used approach, and an improved discussion of our results and we address this in the revised version of our manuscript. With these revisions we address all the specific comments made by the reviewer. Besides this, we try to clarify that the aim of our study is to find general relationships between the spatial heterogeneity of the aquifer matrix and the dynamics of biogeochemical reactions. We do use data from our subject site as well as from the literature to constrain the model scenarios but it has not been our aim to provide a simulation of any specific part of the subject site. For the sake of generality we thus also consider higher initial reaction rates than those likely to be found at the site which allowed us to span a larger range of conditions. We are also fully aware of the uncertainties associated with the parameter values used in our simulations. However the parameter values we used are consistent in their relative magnitude to each other and with the range of experimental observations. We found independently of the compound and its specific reactivity in a given domain that the Damköhler number is a good predictor of the importance of spatial heterogeneities which allowed us to present and discuss our findings independently of the specific parameter values used. In the revised manuscript we expand our discussion of the uncertainties associated with our approach and the associate limitations.*

Response to specific comments:

P1-L18: 'undertook'
*Addressed:*
"…
 Therefore, we undertook a numerical modelling approach to evaluate the sensitivity of groundwater microbial biomass distribution and nutrient cycling to spatial heterogeneity in different scenarios accounting for various residence times.
…"

P1-L21: In biology we have a clear nomenclature. Conditions are either 'oxic' or 'anoxic', Organisms and processes are 'aerobic' or 'anaerobic'. I suggest to use this nomenclature concisely throughout the MS.
*This is exactly how we intended to use it as well. Thus, oxic and anoxic zones refers to presence or absence of oxygen, while aerobic and anaerobic zones refer to the predominant respiration type that is*

*occurring at that location. We can clarify this wording further in the revised manuscript, and we have*
*proof-read the manuscript to ensure that we do not misuse the terminology. Note that we have used the term "oxic/anoxic cell" when referring to grid cells of the numerical model. We replace this term in the revised manuscript to avoid any confusion: (revised to "mesh nodes").*

P2-L37: Here and at many other spots
*We assume that this comment pertains to the references shared by the referee. If not, could they please clarify the specific comment associated with this line?*

P2-L49: Papers of potential interest for the authors: Zhou et al. (2012) FEMS Microbiol. Ecol. 81: 230-242, Hofmann et al. (2020) Front. Microbiol. 11: 543567
*Both the papers are interesting and are now referred to in the Introduction section.*
"...
Schwab et al. (2017);Zhou et al. (2012);Hofmann et al. (2020) linked changing diversity of microbial communities in groundwater with spatio-temporal variation of the groundwater 50 physico-chemical quality.
…"

P2-L50: Papers of potential interest for the authors: McGuire et al. (2000) Chem Geol 169: 471–485, McGuire et al. (2005) Ground Water 43: 518–530

*These papers suitably display the point that physicochemical characteristics of and redox zones in*
*groundwater vary in both space and time (although for contaminated sites and not for oligotrophic conditions as in our study) and are referred to now in the Introduction section.*
"...
McGuire et al. (2000) and Benk et al. (2019);McGuire et al. (2000) linked changing composition of terminal acceptors and of dissolved organic matter (DOM) in groundwater with surficial events,
respectively.
…"

In the Introduction section important issues such as the discrimination between 'active' and 'inactive' as well as 'mobile' and 'immobile' cells are not picked out as central points.
*The central point of this work is to enhance understanding about how spatial heterogeneity controls access to nutrients and in turn shapes the microbial community in a system that provides feedback on carbon and nitrogen cycling. We thus did not focus on the discrimination between different types of microbial cell in the introduction. However, following the suggestion of the reviewer we now address this issue in the Introduction).*
"...
Spatial heterogeneity influences subsurface microbial and nutrient dynamics by limiting access to nutrients and electron acceptors (Murphy et al., 1997), thereby influencing the distribution of active, dormant, suspended and attached microbes as well (Grösbacher et al., 2018; Couradeau et al. 2019). Dormant microbes were found to vary between 60% to 80% of total microbial 70 biomass in soil (Lennon
and Jones, 2011), and attached microbes commonly form the majority of microbial biomass fraction in the subsurface (Griebler and Lueders, 2009; Grösbacher et al., 2018). However, data on these fractions for groundwater systems are still scarce.
..."

Methods
As highlighted in the first paragraph, the conceptual model is simplistic. While the focus is to test for spatial heterogeneity and flow velocity as steering factors with respect to carbon, nutrient and microbial dynamics, frame conditions for the model simulations are non-dynamic with steady-state flow and constant inflow concentrations of dissolved species. In this respect, I would expect an in-depth discussion of the model output. Are the set frame conditions sensitive factors? How may the results change with respect to transient input concentrations of carbon and nutrients.

*We fully realise that we are not exploring all the factors that govern microbial activity and nutrient cycling in a system. The study is limited to studying the impact of spatial heterogeneity and how it*
*controls access to carbon substrate and electron donors and acceptors. We compensate for the lack of exploring other factors (such as varying concentrations of chemical inputs) by varying the flow regime. For example, a higher substrate input at the inlet in a slow flow regime will result in a reaction dominant regime (recall that the slow flow regime is already reaction dominant). With the consideration of a variety of electron acceptors in varying flow regimes we do cover a comprehensive set of conditions for*
*the research question that is explored in the manuscript. As suggested by the referee, the impact of temporal dynamics, or transient conditions, is an interesting and pertinent question in the field. But it results in more confounding factors, and thus we believe that we won't be able to do justice to the question at hand, i.e., how does spatial heterogeneity impact microbial activity and nutrient cycling in a system? We now clarify the aim of our approach at several locations in the Introduction:*
"...
Since preferential flow paths have been established to control access to nutrients, electron acceptors and thus influence the emergence of microbial hotspots (Franklin et al., 2019), we focus on investigating spatial heterogeneity alone.
…"
"…
Simulated scenarios are informed by data from the literature and from a subject site to describe realistic although generic conditions, which allows us to combine these conditions with different types of subsurface heterogeneities to determine the resulting biogeochemical potential of the subsurface system.
…"
*And in the Methods sections:*
"…
It is however not the aim to explicitly simulate and specific part of the subject site. For some model input we rather considered values at the extreme end of possible conditions to enlarge the range of conditions covered by our model scenarios.
…"
*We also discuss its limitation and suggest further research steps building upon the result of our study in the Discussion section:*
"...
The reaction network was formulated using literature knowledge and geomicrobial activity identified at
the subject site. At the same time, it captures varying respiration and microbial regimes, from aerobic autotrophy to aerobic heterotrophy and anaerobic heterotrophy. The activity of geomicrobial reactive systems is dependent on a variety of factors, such as nutrient availability, access to energy gradients, pH, pore size, hydraulic conductivity, particle size distribution (Smith et al., 2018). The limited information on 490 microbial activity applicable to oligotrophic conditions in the subsurface does challenge the
parameterization of the reaction network, which is a priori a potential major source of uncertainty for the obtained model results. Given this limitation, we calibrated the parameters of the reaction network to ensure that it covers a sufficiently large range of Da values and that it does not violate the established redox hierarchy in any of the flow regimes considered (see Appendix A and the base case results). Additionally, 495 we consistently used our parameter set in all scenarios and used results of the
homogeneous base cases as internal reference to which we compared results of the individual heterogeneous scenarios as we aimed to study the impact of spatial heterogeneity on microbial activity and subsurficial nutrient dynamics.
Lastly, consideration of varying flow regimes in combination with the reaction network provides a view on both reaction dominant systems and flow dominant systems, indicated by the use of Da. This approach
500 compensates for our approach wherein we do not explore additional scenarios varying concentrations of chemical species and their influence on microbial growth and distribution. By treating the analysis of results in terms of Da, we condense the discussion to effective rates of microbial activity given presence of spatial heterogeneity of hydraulic conductivity.

*...”*

*“...*

We expect additional studies exploring the impact of varying concentrations of chemical species, parameters relevant to other ecosystems or subject sites to add to the evidence generated by our study that the impact of spatial heterogeneity on subsurficial reactive systems may be predicted using field estimated indicators such as breakthrough time, Pe and Da.

…”

The authors mention the 'use of geochemical and geomicrobial observations from a common study site' as basis of the conceptual model. However, I could not find any sources (papers cited) with respect to 'values'. The concentrations of TOC, DOC, NH4,

NO3, O2, prokaryotic cells (active and inactive) have been selected and based on which studies and sites.).

*In P6-L176, we refer to Kuesel et al. (2016) that presents concentrations of DOC, TOC, nitrate, DO (Figure 6) observed in the groundwater at the subject site. We used the upper limits of observations made for the reactive species at the site, as clarified in the Methods section:*

*“...*

It is however not the aim to explicitly simulate and specific part of the subject site. For some model input we rather considered values at the extreme end of possible conditions to enlarge the range of conditions covered by our model scenarios.

…”

*The paper also mentions the geomicrobial activity that has been identified at the site: aerobic and anaerobic heterotrophy as well as autotrophy, and we duly noted them and included them in the process network. At the same time, we made sure to calibrate the process network so that it adheres to redox hierarchy (see Appendix A for details on the process network). This also enabled us to generalize the findings.*

In P6-L182 you say: 'The concentrations of the reactive species mimicked conditions observed in the subject site'. However, in the discussion it is mentioned that there were two orders of magnitude difference in prokaryotic cell numbers. I ask the authors to carefully consider input values. Is it true that a prokaryotic concentration of mobile prokaryotic cells in groundwater of 10^9 have been found in the field? Seems very high to me.

*It looks like there is confusion with respect to the use of reactive species. By reactive species, we refer to DOC, ammonium, nitrate etc. (not the microbial species). Naturally, having used data collected at the site, we drew a comparison between the microbial biomass present at the site and in our domain.*

*However, it must be kept in mind that our scenarios are not meant to resemble exactly the conditions at specific locations of the subject site. We defined our input concentrations by the upper limits of the concentrations having been measured at the subject site which rather represent conditions close to the groundwater table. This allowed us to consider a wider range of Damköhler numbers including high reactivities. This explains that also the bacterial number are higher than observed at the well of the*

*subject side with travel times of years to decades needed for the recharged groundwater to reach them. If we would have considered conditions of low reactivity/Damköhler numbers, matching conditions around the observation wells, a better match of the observed bacterial cell numbers would have been possible. But answering the question of the study, how spatial heterogeneities affect the dynamics of biogeochemical transformations at a variety of conditions found in groundwater, would not have been*

*possible. This was the motivation to also bring into perspective findings from other studies and sites  and we included the above points into the discussion of the revised manuscript:*

*"...*
However, the simulated values of mobile biomass are in the range derived in both lab scale and field scale studies (Holm et al., 1992;Griebler and Lueders, 2009;Grösbacher et al., 2018). Also, the mobile biomass concentration is in the range of particulate organic carbon concentration observed to be exported in the seepage at the subject site (Lehmann et al., 2021). The relatively high biomass values obtained in the 515 simulations are attributed to the relatively high inflow concentrations as well as to the relatively high microbial reactivity we considered in the simulations to allow them to cover also high Da ranges. We note that while the total biomass may not be matching the observations at the subject site, the relative composition of the microbial species fractions (that is, immobile, mobile, active and dormant) follow established findings. For example, immobile microbial biomass indeed forms the majority biomass in the 520 subsurface (as well as in our study), with its ratio with mobile biomass changing based on nutrient and other environmental conditions (Griebler et al., 2002; Grösbacher et al., 2018). It is proposed that the ratio of immobile and mobile biomass in (Griebler et al., 2002; Grösbacher et al., 2018) varies per nutrient availability, with higher ratios observed in oligotrophic conditions and lower ratios in nutrient rich conditions. We extend this further in our study, by observing that the ratio depends on the Damköhler 525 number, with higher ratios in in low Da systems, and lower ratios in high Da (reaction dominant) systems. It is further estimated that 60%-80% of microbial biomass in soil may be inactive (Lennon and Jones, 2012). In our study, we observe these ranges in the slow flow and medium flow regimes, but not in the fast flow regime. With newer technologies equipped to better characterise activity of microbes in environmental samples (Couradeau et al. 2019), we expect that it will be easier to draw the comparison in the future.
*…"*

*Having said that, we acknowledge that the gene count in Opitz et al. (2014) is actually $10^6$ to $10^8$, we have corrected this in the revised version:*
*"...*
Opitz et al. (2014) measured the total bacterial biomass in groundwater of the subject site to vary from $10_6$ to $10_8$ gene copies $L_{-1}$ (depending on 510 location, tapped aquifer and season of measurement), which is lower than the simulated mobile values.
*…."*

The concept of the reaction network is simplistic, which is ok. But it should be as realistic as possible. Does all NH4 originate from DOM? Isn't there direct input of NH4 from the surface?
*As clarified earlier, we used the concentration values across the entire site to be the input at the inlet. The concentration of ammonium at the inlet is not attributable solely to the depolymerisation of organic matter (POM), but also attributable to incoming flux from the upstream zones.*

As mentioned in Table A.4.2. there is a constant input of 60µM of NH4 (about 1mg/L) that cannot originate from the 10mg of DOC (800 µM). I recall that the studied Hainich Critical Zone sites are partially located in areas with agriculture. To which of the Hainich sites does the conceptual model refer to?
*The referee understands correctly that the input into soil will be different in different land use areas. At the same time, soil and the vadose zone are good buffer zones. Additionally, the groundwater in the agricultural area lies in a confined aquifer with more limited interaction with the surface. It must be noted that considering the size of our domain, this modelling exercise is not intended to replicate the processes in the entire study site but to understand if there is an impact on how we predict the nutrient cycling in the subsurface were we to neglect spatial heterogeneity. Thus, all results and associated discussions are carried out with respect to the homogeneous domain (a common assumption) being the base case.*

I know that it is hard to collect reliable information from the literature with respect to microbial features in shallow aquifers. Having this in mind, one need to carefully select values for 'rate constants', 'yield coefficients', … The values summarized in Table A.4.1
originate from field studies and lab studies very different in nature, i.e. values derived from lab experiments with pure bacterial cultures. Are the chosen values sufficiently representative for the Critical Zone in Hainich and shallow aquifers in general? This at least needs to be critically discussed.

*We agree that different literature source have been used for the parameter values used in this study we*
*also agree that there is a large uncertainty regarding these values and to which extent they are representative for our subject site. However, we took care that the parameter values are in reasonable relation to each other (i.e. aerobic processes taking place faster and providing higher yields than anaerobic processes etc.). Since rate parameter for DOC degradation are also highly depending on the composition/reactivity of DOC which is highly variable in the subsurface even the use of field derived*
*rate parameters from the subject site or any other field site would not have been representative for shallow aquifers in general. As pointed out above in the context of the bacterial numbers the maximum reaction rates we consider in our study might be at the higher end of the range of rates one would potentially observe at the subject site but this allowed us to cover a larger range of Damköhler numbers when investigating the impact of spatial heterogeneities on biogeochemical transformations, which was*
*our focus in this study. In the revised version we expanded our discussion to address these issues:*
"…

The reaction network was formulated using literature knowledge and geomicrobial activity identified at the subject site. At the same time, it captures varying respiration and microbial regimes, from aerobic autotrophy to aerobic heterotrophy and anaerobic heterotrophy. The activity of geomicrobial reactive
systems is dependent on a variety of factors, such as nutrient availability, access to energy gradients, pH, pore size, hydraulic conductivity, particle size distribution (Smith et al., 2018). The limited information on 490 microbial activity applicable to oligotrophic conditions in the subsurface does challenge the parameterization of the reaction network, which is a priori a potential major source of uncertainty for the obtained model results. Given this limitation, we calibrated the parameters of the reaction network to
ensure that it covers a sufficiently large range of Da values and that it does not violate the established redox hierarchy in any of the flow regimes considered (see Appendix A and the base case results). Additionally, 495 we consistently used our parameter set in all scenarios and used results of the homogeneous base cases as internal reference to which we compared results of the individual heterogeneous scenarios as we aimed to study the impact of spatial heterogeneity on microbial activity
and subsurficial nutrient dynamics.

Lastly, consideration of varying flow regimes in combination with the reaction network provides a view on both reaction dominant systems and flow dominant systems, indicated by the use of Da.
…"

With respect to DOC, an contstant input concentration of 800μM has been chosen. DOC degradation in soil and in groundwater is determined not only by its concentration but more likely by its quality (degradability). Has this been considered.

There is dynamics in many aspects, including flow velocity, water retention time, activity and biomass of microbes, DOM concentration and transformation, N transformation, …
Only a subset of parameters, i.e. spatial heterogeneity and flow velocity (related to residence time) has been tested. This needs to be clearly mentioned already in the Introduction section.

*Clarification added in the introduction section of the revised manuscript:*
"…
Since preferential flow paths have been established to control access to nutrients, electron acceptors and thus influence the emergence of microbial hotspots (Franklin et al., 2019), we focus on investigating spatial heterogeneity alone.
…"

Captions of Tables are generally on top of the tables, not below. See all tables.
      *Noted, corrected in the revised version.*

      P11-L297: Is there evidence for a parallel reduction of nitrate and oxidation of
      ammonium?
*Both ammonia oxidation and nitrate reduction have been found to be active in sub-oxic conditions: (De
      Brabandere et al., 2014; Kalvelage et al., 2013; Seitzinger et al., 2006.) In our simulations these
      processes thus take place sequentially and not in parallel. In particular, in the base cases in our study,
      the overlap of nitrate removal and ammonium removal zones (i.e. the local co-occurrence of both
      processes) is very small, ranging from ~1cm in the base case of the slow flow regime to ~8cm in the base*
*case of the fast flow regime, and characterized by relatively low rates for both processes.*

      P12-L321: Is there any evidence that the portions of active and inactive cells/species are
      realistic? In particular when these ratios are calculated for individual physiological guilds
      (nitrate reducers, ammonium oxidizers, …). See also table 4.
*To the best of our knowledge '…little is known about the relevance and extent of dormancy in
      groundwater systems' (review of Ruiz-Gonzales et al., 2021)  but it is assumed that dormancy is a crucial
      factor for groundwater bacterial communities. For soil systems up to 96% of the bacterial cells we found
      to be inactive (review of Lennon and Jones, 2011) which implies that the numbers obtained in our model
      scenarios are realistic. In the revised manuscript we added a few word on this issue to the Discussion:*
*"…*
      It is further estimated that 60%-80% of microbial biomass in soil may be inactive (Lennon and Jones,
      2012). In our study, we observe these ranges in the slow flow and medium flow regimes, but not in the
      fast flow regime.
      …"
      P15-L386: log10Da?
      *Correct. Thank you for pointing it out. Addressed in the revised version.*

      P15-L394: Dissolved oxygen (DO) is not a nutrient.
*Correct. Thank you for pointing it out. Phrase replaced with "chemical species" in the revised version.*

      P16-L411: Provide a citation that supports this statement.
      *Addressed:*
      "…
This may be due to the presence of DO at reduced concentrations in the downgradient region of the
      domain. DO at these concentrations and low DO/Ammonium ratios can be 425 preferentially taken up by
      ammonia oxidizers compared to aerobic degraders (Gu et al., 2006).
      …"

Discussion

      P19-L459: The 'available' process knowledge, does it refer to the Hainich study site?
      *This phrase refers to knowledge from literature from deep subsurface studies (groundwater, marine
      sediments), and thus includes but is not limited to the Hainich study site. We clarified this in the revised*
*manuscript:*
      "…
      In this study we synthesized available process knowledge and observations from our subject site on
      geomicrobial activity in the deep subsurface, both terrestrial and marine, into a set of in silico scenarios
      on 475 the fate of biogeochemically reactive compounds in heterogeneous subsurface settings.

…"

        P19-L474: from carbon concentration and carbon content per cell one will not end up with
        gene copies per volume but cells per volume.
        *Correct. Thank you for pointing it out, addressed in the revised version of the manuscript:*
*"...*
        Microbial abundance can be derived from carbon content in the biomass using available conversion
        factors varying from 5 - 39 femtogram (fg) C/cell (Fukuda et al., 1998;Vrede et al., 2002). This resulted
        in median values of total mobile biomass in the domain of 10^9 to 10^11 cells/L.
        *…"*
        Quantification of cell numbers in
        groundwater and aquifers by means of molecular tools quantifying 16S rDNA gene copies
        is only a rough estimation of cell abundance.
        *Absolutely, we agree. This contributes to the general uncertainty when comparing the model derived*
*biomass data with experimental data. See also our comment above.*

        P19-L474: Does 100times less cells equal 100times less microbial activity and 100times
        less transformation of C and N? Please comment on that.
        *The overall activity is the product of the number of active cells and the rate per cell at the given*
*conditions. Since the in situ rates per cell are not known for the subject site it is difficult to speculate how*
        *much less activity comes with the reduced cell numbers. As discussed above and in the revised manuscript*
        *the conditions we imposed in the model are more representative for conditions close to the groundwater*
        *table and allow (on purpose) a higher reactivity than in deeper groundwater regions next to the*
        *observation wells.*
        P20-L490 & L516: There is techniques and reports available on high-resolution sampling in
        aquifers. The ready should not get the impression one cannot get spatially more resolved
        in sampling. E.g. Ronen et al. 1987 J. Hydrol. 92, 173–178, Báez-Cazull et al. 2007 Appl.
        Geochem. 22, 2664–2683, Smith et al. 1991Contam. Hydrol. 7, 285–300, Anneser et al.
2008 Appl Geochem 23:1715–1730.
        *Absolutely, as we also mention, we have **some** capability to resolve the heterogeneity and obtain higher*
        *resolution samples, especially in groundwater. But it must be recognised that there will always be sub-*
        *sampling scale heterogeneities that the sampling technique will not be able to resolve. In the revised*
        *manuscript we modified our wording to clarify this point and expanded the references to do the same.*

        *"...*
        The requirement of vertically discretized sampling has already been recognized (Ronen et al., 1987;
        Smith et al., 2018) and addressed by various sampling methodologies such as low flow sampling
techniques, passive samplers, point and discrete interval samplers (Ronen et al., 1987;Smith et al.,
        1991;Powell and Puls, 1993;Báez-Cazull et al., 2007;Anneser et al., 2008) even though sub-sampling
        scale heterogeneities will not be resolved. Our results support the usefulness of such spatially resolved
        575 sampling techniques for analysis of microbial activity in the groundwater.
        *…"*
P20-L501: Have the authors considered that there is different growth rates with different
        physiological groups within the microbes, i.e. aerobes may grow faster that nitrate
        reducers and sulfate reduces are extremely slow. Did I miss this information?
        *The parameterisation of the reaction network addresses this aspect (among many others) of low*
        *respiration rates, low yield coefficients etc (see Table A.4.1). Thus, we see aerobic degraders*
*(heterotrophs) outcompeting other species in presence of DOC and DO, while nitrate reducers become*
        *active only at much lower concentrations of DO (see figures S1-S4).*

P20-Fig. 6: What do you mean with 'oxic cells'. Please change.
*Apologies for the confusion. We refer to the grid cells (0.01mx0.01m) that are oxic at a particular cross-section along the predominant flow path (P20-L495). We replaced this phrase with "mesh nodes" in the revised manuscript to avoid such confusion.*

P21-L528: Consider the review paper of Smith et al. 2018 FEMS Microb. Ecol. 94: fiy191
*Thank you the suggestion. We make reference to this paper in the revised manuscript at several locations including here:*
"...
Immobile microbes account for more microbial activity compared to mobile microbes. However, groundwater samples represent mobile microbial biomass, termed as planktonic biomass (Smith et al., 2018). Estimates of microbial respiration 585 are thereafter made based on the abundance of mobile microbes in the obtained groundwater samples.
…"

P22-L565: I fully agree with this statement. In many cases the contribution of the mobile fraction of microbes can be neglected in terms of 'transformation processes'.
Findings from other studies (like the one already cited Grösbacher et al. 2018) are not discussed in comparison to the model outcome.
*We added some reference to the revised manuscript and discuss our findings in the context of these references:*
"...
The relatively high biomass values obtained in the 515 simulations are attributed to the relatively high inflow concentrations as well as to the relatively high microbial reactivity we considered in the simulations to allow them to cover also high Da ranges. We note that while the total biomass may not be matching the observations at the subject site, the relative composition of the microbial species fractions (that is, immobile, mobile, active and dormant) follow established findings. For example, immobile microbial biomass indeed forms the majority biomass in the 520 subsurface (as well as in our study), with its ratio with mobile biomass changing based on nutrient and other environmental conditions (Griebler et al., 2002; Grösbacher et al., 2018). It is proposed that the ratio of immobile and mobile biomass in (Griebler et al., 2002; Grösbacher et al., 2018) varies per nutrient availability, with higher ratios observed in oligotrophic conditions and lower ratios in nutrient rich conditions. We extend this further in our study, by observing that the ratio depends on the Damköhler 525 number, with higher ratios in in low Da systems, and lower ratios in high Da (reaction dominant) systems.
…"
"…
This indicates that the mobile microbial abundance detected in groundwater samples must be used with care as a proxy for effective microbial activity and nutrient cycling (also confirmed by Alfreider et al. (1997), Murphy et al., (1997), Griebler et al. (2002) and Grösbacher et al. (2018) as mentioned earlier).
…"

Summary and conclusion

P24-L630: mention at which spatial scale.
*Addressed in the revised manuscript:*
"...
In this study, we investigated the impact of spatial heterogeneity on biomass persistence, distribution, and nutrient cycling at the sub-meter scale in the subsurface.
…"

P24-L640: Can this be visualized?

*We display this in select heterogeneous scenarios in Fig S2 (1D spatially averaged concentration of different microbial species) and Fig S4 (2D concentration heatmaps of different microbial species). In addition, we display how predominantly anaerobic cross-sections in the domain may still contain oxic niches in Fig 6.*

---

## Author Comment (AC2)

**Predicting the impact of spatial heterogeneity on microbial redox dynamics and nutrient cycling in the subsurface**

Swamini Khurana[1], Falk Heße[2,3], Anke Hildebrandt[2,4,5], Martin Thullner[1]

[1]Department of Environmental Microbiology, Helmholtz Centre for Environmental Research – UFZ, Leipzig, 04318, Germany
[2]Department of Computational Hydrosystems, Helmholtz Centre for Environmental Research – UFZ, Leipzig, 04318, Germany
[3]Institute of Earth and Environmental Sciences, University Potsdam, Potsdam, Germany
[4]Institute of Geoscience, Friedrich-Schiller-University Jena, Jena, Germany
[5]German Centre for Integrative Biodiversity Research, Leipzig, Germany

**Correspondence**: Swamini Khurana (swamini.khurana@ufz.de)

**Response to RC2**

*We thank the reviewer for the positive feedback with respect to the relevance of the work for the readership of the journal, and for the constructive comments to make the work more accessible. We acknowledge that we have to provide an improved discussion with respect to the applicability of the results for real world systems and also at larger scales. We will update the manuscript addressing this gap, as well as addressing the specific comments made by the reviewer below.*

L1: The term "redox dynamics" is not used in the manuscript (except once when referring to the literature) and I am not entirely sure what the authors want to convey with it.

*We agree with the reviewer and propose updating the title to the following:*
*"Predicting the impact of spatial heterogeneity on microbially mediated nutrient cycling in the subsurface"*

L58: "in this microbial ecosystems...": what does "in this" refer to?
*'this' refers to biogeochemical cycles. We agree that the phrasing is ambiguous. We will rephrase the sentence as given below:*
*"...In these biogeochemical cycles, microbial communities play a key role …"*

L78: "Sufficiently well" for what?
*We apologize for the ambiguous qualifier. We will rephrase this sentence as follows:*

*"Biogeochemical reaction networks have been explored extensively over the past decades with improvement…."*

L81: Please add citations for the statement on biogeochemical reaction networks.
*Addressed.*

L81-82: It is not clear to me how the sentence starting with "Working with ..." fits into the line of arguments here.

*We agree that the phrasing is ambiguous. Here, we attempted to describe how biogeochemical reaction networks improved over the years, from only physical processes describing element fluxes to moving towards microbial explicit models. We have now rephrased:*

*"Incorporating microbially explicit reaction networks in reactive transport models …."*

L84: "A straight-forward application of the soil-based biogeochemical model approaches to conditions in deeper subsurface compartments is problematic because the nature of carbon source changes as it travels into the deeper zones." I believe the authors did not specifically look into this- is there a reson for this?

*The reviewer is correct in that we didn't go into a detailed characterization of dissolved organic carbon in the deep subsurface as this was beyond the scope of the study, and studies such as Benk et al (2019) already explore this aspect. This section, on the other hand, primarily motivated the development and parameterization of a reaction network that is appropriate for deep subsurface oligotrophic environments. To develop a reaction network for the deep subsurface, we adapted conceptual approaches from numerous studies (L151-L163), and reparametrized the reaction network (described in Appendix A) which took into account slow reactivity of DOC for example.*

L99: I think using the term "mechanisms" is not great here as I think the manuscript does not address this.

*We acknowledge that the manuscript does not specifically address the mechanistic understanding microbial nutrient dynamics in the subsurface. This section primarily described the shortcoming of field scale studies, in that they do not provide insights into the mechanisms governing microbially mediated nutrient cycling in the subsurface. This statement, even though not addressed in the paper, is a drawback of field scale studies. This is overcome-to an extent-by pore-scale studies that we describe subsequently. Although we think that using "mechanisms" helps in the flow of argumentation of the text, we will remove it since the reviewer thinks otherwise.*

L196: "established": can you add references for this?
*We understand that the reviewer is concerned about using "established" as a qualifier. While variograms have been used and discussed in numerous works: (Gelhar and Axness, 1983;Johnson and Dreiss, 1989;Webb and Andersen, 1996, Berkowitz, 2002;Dagan et al., 2003;Delhomme, 1979;Heße et al., 2014), we propose to remove the qualifier as it may lead to some confusion. We will thus rephrase the sentence as follows:*
*"… To conceptualize heterogeneity, we used a limited parameter set, i.e., variance in the log normal distribution …."*

L196: What are variance and anisotropy values used for the base case? When I first looked at Figs. S1 and S2, I was confused because the homogeneous base case was shown in all variance:anisotropy ratios.
*Homogeneous domain is the base case. Homogeneous domains are characterized by the same value of conductivity throughout the domain. Whereas variance and anisotropy are characteristics of heterogeneously distributed properties (hydraulic conductivity in our case). So, the base case (homogeneous domains) is characterized by the same value of hydraulic conductivity at each node (variance is 0, and there is no associated anisotropy).*
*We recognize the source of confusion as we have not explicitly stated this in the manuscript. To rectify this, we will revise the first entry in Table 2 (where the first row refers to the homogeneous domain). We will also update the caption of Fig. S1 and Fig S2:*

"

*Figure S1: Flux averaged concentrations of dissolved species in heterogeneous domains (indicated by variance and anisotropy values in the row index) in three types of heterogeneous scenarios (solid lines) compared to that in the homogeneous base case (zero variance and no associated anisotropy, dashed-dot lines) in all flow regimes. The flux averaged concentration profile is the same for a given column (i.e., there is only one homogeneous/base case for comparison in each flow regime).*

*Figure S2: Spatially averaged concentration profile of the immobile active biomass in heterogeneous domains (indicated by variance and anisotropy values in the row index) in three types of heterogeneous scenarios (solid lines) compared to that in the homogeneous base case (zero variance and no associated anisotropy, dashed-dot lines) in all flow regimes. The spatially averaged concentration profile is the same for a given column (i.e., there is only one homogeneous/base case for comparison in each flow regime).*
"

L250: Based on equation 2, the Da value should depend on the size of the domain relative to the size of the "heterogeneity". Have the authors looked into this?
*We understand that there is some confusion regarding the calculation of Da in each systems. We have expanded the section describing its calculation. We hope that it is more comprehensible and easier for readers to adapt for their own studies as well.*

L312: "while the removal of TOC was the lowest there...": is this trend related to microbial biomass?
*Correct. Please refer to Fig. S9 where we display that inactive mobile biomass contributes substantially to the total biomass, and it is considered in calculation of TOC (L238:241).*

L391-392: The bars in Fig. S6 are not linked to redox conditions- is it possible to do so?
*We assume that the reviewer sis looking for a chemical species-specific distribution of Damköhler number like given below? We will add this as additional figure to the SI since we do not see a way to provide all information in one figure.*

[Figure]

L441-451: Given that the prediction of the impact of spatial heterogeneity on redox

regimes is the major posit of this manuscript, I believe this section needs to be improved. A few suggestions for improvement are given in the following few comments.

L444: What is AIC and what do these values mean?

We refer to AIC in the Methods section (L245) as the Akaike Information Criterion (L245). It is an indicator of prediction error of a general linear model. It is a relative criterion commonly used to compare the performance of a collection of models. The model that has a lower value of AIC performs better. We recognize the requirement to explain it better and so we updated the Methods section (L245) with further information on it:

*"We compared the Akaike Information Criterion (AIC) of each model to evaluate the fit of the model. AIC is an indicator of prediction error associated with a general linear model. It is an indicator of relative performance of a group of models; the model with the lowest AIC is concluded to be the one with least prediction error or best performance. With each iteration of the model, we selected the features most influencing the performance of the model and reducing the AIC of the predictions...."*

*Additionally, we updated Section 3.5 to explain the results in a better way:*

*"*

*While conducting the multivariate statistical analysis of change in mass removal of reactive species, we made use of AIC to evaluate governing factors influencing mass removal in a spatially heterogeneous domain. The analysis indicated that AIC was 994 when considering only breakthrough time and chemical species. AIC reduced to -211 when the chemical species, the flow regime, variance in permeability field and the anisotropy of the domain were included as random factors). Please refer to Table S1 for further details. Thus, we concluded that nutrient dynamics is influenced by spatial heterogeneity. Categorizing the systems using log10Da, we proposed a linear expression to predict the impact of spatial heterogeneity on nutrient removal. The regression parameters informing this expression are given in Table 5. The results indicated that we may underestimate nutrient removal by 6 times or overestimate it by twice the amount (Fig. 5).*
*"*

L449: Where in Fig. 5 can I see these under-/overestimations?
*We agree that the caption of Figure 5 can be vastly improved to equip the reader to match the visualization with the text. The Y-axis in Fig. 5 displays these under-overestimations. We will rephrase the figure caption as follows. We hope that this explains the axes and the data referred to in the text better.*
*"Figure 5: Regression analysis: Predicting impact of spatial heterogeneity on chemical species removal in different reaction regimes indicated by log10Da. Value on Y-axis indicate the removal of chemical species in heterogeneous domains normalized by that in the corresponding base case. Spatial heterogeneity is plotted on the X-axis, indicated by the breakthrough time in the heterogeneous domain normalized by that in the base case (homogeneous domain). A value of 100% on the Y-axis indicates that the removal of the chemical species is the same as that in the corresponding base case (homogeneous domain). A value of 50% indicates that the removal of the chemical species reduced by half in the corresponding heterogeneous domain. A value of 600 indicates that the removal of the chemical species in the heterogeneous domain was 6 times that in the homogeneous domain."*

L452: Does this Fig. include data for different reactive species? Also, why are Da numbers given in log10 base (given that per definition Da is already the ln of the concentration ratio between outflow and inflow)? If solutes are consumed in the domain, then Cout/Cin is always < 1 and hence the Da per eq. 2 is negative- which will give a complex number when taking log10. Is there something I am missing here?

*We understand that it was confusing to follow the calculation of Da. So we updated the Methods section with the following details. We also acknowledge that there was an error in the formula presented for calculating the Da earlier ( a factor of -1 was missing); it is now corrected in the updated section:*
*"*

*we used the Damköhler number (Da) to indicate the reaction regime for each reactive species. Da is defined as the ratio of the advective transport time scale and the reaction time scale as described in Eq. 2.*

$$Da = \frac{\tau_{transport}}{\tau_{reaction}}, \qquad\qquad (2)$$

*where, τreaction is the characteristic reaction time scale and τtransport is the characteristic transport time scale given by the breakthrough time of a conservative tracer in the domain. We adapted this definition and used Eq 3 below to calculate the apparent Da using values estimable in the field when* $\frac{C_{out}}{C_{in}} > 5\%.$

$$Da = -ln\frac{C_{out}}{C_{in}}, \qquad\qquad (3)$$

*with Cin as flux averaged concentration of a reactive species entering the domain, and Cout as flux averaged concentration of the reactive species leaving the domain. In case of* $\frac{C_{out}}{C_{in}} \leq 5\%$, *we used Eq. 4 and Eq. 5 to derive the apparent Da of the chemical species*

$$\tau_{reaction} = \frac{-ln\,(0.37)}{-ln(\frac{C_{y5}}{C_{in}})} \times \tau_{y5}, \qquad\qquad (4)$$

$$\tau_{reaction} = \frac{\tau_{y5}}{ln(\frac{C_{y5}}{C_{in}})}, \qquad\qquad (5)$$

*where,* $C_{y5}$ *is the concentration of the chemical species at the first cross-section (y = y5) when* $\frac{C}{C_{in}} \leq 5\%,$ *and* $\tau_{y5}$ *is the breakthrough time for a conservative tracer at the same cross-section, i.e., y = y5.* $\tau_{transport}$ *in this case was the same as the breakthrough time of the conservative tracer in the domain (Eq. 6).*

$$Da = \frac{breakthrough\ time}{\frac{\tau_{y5}}{ln(\frac{C_{y5}}{C_{in}})}}, \qquad\qquad (6)$$

*Thus, we were able to characterize reaction dominant system where Da > 1. We took the logarithm of Da to the base 10 (log10Da) to characterize the regime for each reactive species in each domain.*
*For a scalable relationship addressing impact of spatial heterogeneity on reactive species removal, we conduct a simple linear regression analysis of species removal vs. residence time (both in relative units to the homogeneous reference cases) for different log10Da ranges.*
*"*

L461: I think it would be beneficial to explicitly state the range of the scenarios.
*We agree with the reviewer that it will be beneficial to refer to the range of scenarios that we have described earlier in the text. We will revise the sentence as below:*

*" ...This approach allowed us to generate a wide range of spatially heterogeneous domains (with variance of the log normal distribution of conductivity varying from 0.1 to 10, and anisotropy varying from 2 to 10), which is not possible experimentally. ..."*

L463: "correlation length": this is not discussed in detail in the results section and I am not sure how it fits in here.
*We agree with the reviewer here. Since we didn't vary the correlation length (kept it constant in all the scenarios), we don't need to mention it in this section. We will remove it from the sentence.*

L507-508: "We establish that the persistence of microbial species in the domain is governed by the presence of the appropriate carbon source and electron acceptor, ..." I

am wondering how microbial species are linked to carbon sources and electron acceptors in the model. Is species distribution independently modelled or could this finding in part be a result of the way the reaction network and model are set up?

*In Appendix A, we describe the rate expressions that we used in the reaction network. We adapted Michaelis Menten kinetics for the rate of microbial respiration (section A.3.1.) with carbon substrate concentration (DOC) and electron acceptor concentration (DO for aerobic degraders, nitrate for nitrate reducers and so on). Additionally, we link microbial growth (section A.3.2) with the rate of respiration, ammonium availability and yield coefficient. So microbial biomass is a result of the reaction network.*

*We did not explicitly specify the distribution of either chemical species or microbial species. The species distribution and results thereof that we present are at steady state conditions, which the model itself reaches given uniform species distribution as initial conditions. Thus, the distribution of species evolved due to the spatial heterogeneity of the domain.*

L627: Can the authors elaborate on the significance of this results for environmental systems? For example, when and where do they expect these heterogeneities to be most significant?
*We make a reference to the geological settings which can be represented using the simulated random fields in L464-L467. We further add scenarios in this passage where we expect the results of the studies to be applicable.*
*"*

*We expect advection dominated systems to be impacted by spatial heterogeneity because spatial heterogeneity had a higher impact on the transport profiles in these systems. These are typically systems that are shallow, less compacted (in case of alluvial sediments), or fractured rock systems. Furthermore, the shallow subsurface also receives bioavailable and reactive organic matter with the incoming water which enables a relatively high microbial activity. In contrast, in the deep subsurface microbial activity is lower and rather relies on inputs from the matrix material, which is ubiquitous and doesn't rely on transport for access. We expect additional studies exploring the impact of varying concentrations of chemical species, parameters relevant to these ecosystems or subject sites to add to the evidence generated by our study that the impact of spatial heterogeneity on subsurficial reactive systems may be predicted using field estimated indicators such as breakthrough time, Pe and Da.*
*"*
L654: Or underestimate nutrient removal six-fold as stated in L443?

*Yes, we agree that it may result in underestimation of nutrient removal. We discussed this in detail (L621:624), where we clarify that the 6-fold increase in removal is likely due to the small domain size. The low Da range refers to nitrate removal in the fast flow regime, and nitrate removal in the base case was ~1-2 μM. Thus, a 6-fold increase actually means that the removal increased to 6 μM, which is still low in absolute terms. Thus, we do not make reference to it again in the Summary section (L654).*

Technical corrections:
L18: "used" instead of "undertake"
*Noted.*

L83: "group" instead of "groups"
*Noted.*

L115: "attempted" instead of "attempt"
*Noted.*

L123: "Disentangle" from what? I think describe/define would be better.
*Noted.*

L271: I think the "removal" before "impact" should be deleted.
*Noted.*

L424: Why are some words bold?
*Noted: It was a formatting error.*

L446: Seems like a repetition of L 443f.
*Noted. We propose to update the section as follows:*

*"While conducting the multivariate statistical analysis of change in mass removal of reactive species, we made use of AIC to evaluate governing factors influencing mass removal in a spatially heterogeneous domain. The analysis indicated that AIC was 994 when considering only breakthrough time and chemical species. AIC reduced to -211 when the chemical species, the flow regime, variance in permeability field and the anisotropy of the domain were included as random factors). Please refer to Table S1 for further details. Thus, we concluded that nutrient dynamics is influenced by spatial heterogeneity. Categorizing the systems using log10Da, we proposed a linear expression to predict the impact of spatial heterogeneity on nutrient removal. The regression parameters informing this expression are given in Table 5. The results indicated that we may underestimate nutrient removal by 6 times or overestimate it by twice the amount (Fig. 5)."*

L553-555: A verb is missing in this sentence.
*This is possibly a confusion between "nitrate reducers" and "nitrate reduces". We will rephrase.*
*"It must be noted though that the concentration of nitrate decreases when and where the concentration of DO is below 15 uM (Fig. S1)."*

L609: Delete "towards".
*It looks like there is some confusion with the current phrasing of the sentence. We have rephrased this as follows and we hope this alleviates the confusion.*

*"For regimes where -1<log10Da, 0, first order kinetics may be substituted with zero order kinetics"*

Tables: Captions should be above tables, not below.
*Noted.*

Figures: It would be helpful to have different symbols for the three flow regimes so that the figures are readable in black and white.

*Noted.*

---

## Author Comment (AC3)

**Predicting the impact of spatial heterogeneity on microbial redox dynamics and nutrient cycling in the subsurface**

Swamini Khurana[1], Falk Heße[2,3], Anke Hildebrandt[2,4,5], Martin Thullner[1]

[1]Department of Environmental Microbiology, Helmholtz Centre for Environmental Research – UFZ, Leipzig, 04318, Germany
[2]Department of Computational Hydrosystems, Helmholtz Centre for Environmental Research – UFZ, Leipzig, 04318, Germany
[3]Institute of Earth and Environmental Sciences, University Potsdam, Potsdam, Germany
[4]Institute of Geoscience, Friedrich-Schiller-University Jena, Jena, Germany
[5]German Centre for Integrative Biodiversity Research, Leipzig, Germany

**Correspondence**: Swamini Khurana (swamini.khurana@ufz.de)

**Response to RC3**

**Specific Comments:**
There are a few matters that require clarification for the reader and provision of supporting information that is not in the current manuscript:

Line 170: "average hydraulic conductivity" – what average is used (arithmetic, geometric, harmonic)? I would assume arithmetic since not stated otherwise, but often the geometric mean is considered more representative of the average behavior of a heterogeneous permeability field.

*The reviewer is correct in assuming that it is arithmetic average. In addition, we would like to clarify that we ensured that the average water flux in all the domains in a particular flow regime was the same by adjusting the hydraulic gradient between the inlet and outlet boundaries (L167-169).*

Table 1: The "length scale" is given here as 0.1 meters, apparently as used to compute the Peclet number (ref. line 230). What is that value based on?

*We apologise for the confusion. To calculate the Peclet number, the domain length was used (0.5m) to derive the Peclet number for the flow processes occurring at the observation scale. To avoid confusion, we propose to delete this entry from Table 1.*

Line 195: What is the correlation length used in the simulations? I couldn't find it in the tables.

*We apologise for the confusion. The correlation length referred to in this sentence is the autocorrelation spatial/length scale. Its value is 0.1 m (20% of the domain size considering that several studies have found correlation length to be 10-30% of the observation scale (Turcke and Kueper (1996) found typical correlation length to be varying from 0.16m and 0.23m in core sizes of 1.5m while Welhan and Reed (1997) observed correlation length to be 1.5 km in the total domain size of 15 km)), We will update this sentence to prevent further confusion:*
*" … Each random field was characterized by the same mean value of conductivity (i.e., average conditions at the subject site (Jing et al., 2017)) and spatial autocorrelation length scale (0.1 m) in all realizations, in scaling with the size of the domain in line with previous studies (Turcke and Kuper, 1996; Welhan and Reed, 1997; Desbarats and Bachu, 1994)."*

Lines 196-198: The outcomes may be sensitive to the assumption of a second-order stationary random field with horizontal anisotropy. Other types of heterogeneity (e.g., multipoint statistical models, geometric models, or depositional process models) could lead to different (and probably even more striking) conclusions. I don't view this as a flaw of the study, since this assumption is conventional, but the assumption and its potential implications should perhaps be discussed.

*We agree with the reviewer that other models are available to describe spatial heterogeneity. We selected the variogram method as it condensed the representation of the spatially heterogeneous fields in a limited parameter set (de Marsily et al., 2005). Thus, it was able to represent a wide variety of heterogeneous fields regardless of geological or depositional processes involved that resulted in the creation of such spatially heterogeneous domains in the first place, or without incorporating large datasets (as required in multipoint approaches). In our work, we overcame any bias induced by the variogram approach by using the breakthrough time to indicate the extent of spatial heterogeneity, independent of how heterogeneity is described. This led us to discuss results on impact on nutrient cycling and biomass in terms of reduction in breakthrough time given the same average flow conditions. We propose that the same approach could be followed regardless of the heterogeneity model used to generate such spatial random fields. We will add a note on this aspect in the Discussion section of the revised manuscript.*

Lines 245-246: Clarification is needed here to elucidate what is meant by "fit of the model" in defining the AIC criterion. It isn't clear whether this is fit to actual field observations (there are some discussed in the paper but they are not described in detail) or fit to analytical solutions (e.g. Figure S8), or something else.

*We understand the source of ambiguity, we have updated this section as follows:*
*"*

*To evaluate the key factors determining the impact of spatial heterogeneity on nutrient cycling, we undertook a series of multivariate statistical analyses of the simulation results using Linear Mixed Effect Modelling, progressively including variables in both fixed effects and random effects. We compared the Akaike Information Criterion (AIC) of each model to evaluate the fit of the model. AIC is an indicator of prediction error associated with a general linear model. It is an indicator of relative performance of a group of models; the model with the lowest AIC is concluded to be the one with least prediction error or best performance. With each iteration of the model, we selected the features most influencing the performance of the model and reducing the AIC of the predictions.*
*"*

Line 253 (Equation 2): Is this a standard definition of Da? If so, please provide a citation. If not, please provide clarification as to how this equation represents the ratio of advective and reactive time scales.

*We realize that this equation was not adequate to explain the derivation of Da in the scenarios. We will update the section with the following explanation on calculation of Da for the various scenarios. We also noticed there was an error in the equation which is rectified in the explanation below as well.*
*"*
*we used the Damköhler number (Da) to indicate the reaction regime for each reactive species. Da is defined as the ratio of the advective transport time scale and the reaction time scale as described in Eq. 2.*

$$Da = \frac{\tau_{transport}}{\tau_{reaction}}, \tag{2}$$

*where, $\tau_{reaction}$ is the characteristic reaction time scale and $\tau_{transport}$ is the characteristic transport time scale given by the breakthrough time of a conservative tracer in the domain. We adapted this definition*

*and used Eq 3 below to calculate the apparent Da using values estimable in the field when $\frac{C_{out}}{C_{in}} > 5\%$ (adapted from Pittroff et al., 2017).*

$$Da = -ln\frac{C_{out}}{C_{in}}, \tag{3}$$

*with Cin as flux averaged concentration of a reactive species entering the domain, and Cout as flux averaged concentration of the reactive species leaving the domain. In case of $\frac{C_{out}}{C_{in}} \leq 5\%$, we used Eq. 4 and Eq. 5 to derive the apparent Da of the chemical species*

$$\tau_{reaction} = \frac{-ln\,(0.37)}{-ln(\frac{C_{y5}}{C_{in}})} \times \tau_{y5}, \tag{4}$$

$$\tau_{reaction} = \frac{\tau_{y5}}{ln(\frac{C_{y5}}{C_{in}})}, \tag{5}$$

*where, $C_{y5}$ is the concentration of the chemical species at the first cross-section (y = y5) when $\frac{C}{C_{in}} \leq 5\%$, and $\tau_{y5}$ is the breakthrough time for a conservative tracer at the same cross-section, i.e., y = y5. $\tau_{transport}$ in this case was the same as the breakthrough time of the conservative tracer in the domain (Eq. 6).*

$$Da = \frac{breakthrough\ time}{\frac{\tau_{y5}}{ln(\frac{C_{y5}}{C_{in}})}}, \tag{6}$$

*Thus, we were able to characterize reaction dominant system where Da > 1. We took the logarithm of Da to the base 10 (log₁₀Da) to characterize the regime for each reactive species in each domain.*
*For a scalable relationship addressing impact of spatial heterogeneity on reactive species removal, we conduct a simple linear regression analysis of species removal vs. residence time (both in relative units to the homogeneous reference cases) for different log₁₀Da ranges.*
*"*

Lines 265-270: These lines highlight a broader issue: What times were used for analysis and metrics of reactive processes? The flow field is clearly stated as being steady, but the concentration fields will be transient, and various times are mentioned in the manuscript. The breakthrough time is defined as the time at which Cout/Cin = 0.5 (for tracer), but what other times are considered for reactive species (they may not reach this value at the outlet)? This was confusing to me as a reader and should be clarified.

*We agree that this aspect is confusing for the reader. We will rephrase the Data Analysis section to specify that (with the exception of the tracer tests) we are using the steady state concentration profiles for chemical and microbial species as well. Thus, both flow field and concentration fields are at steady state.*
*L237 onwards update to the text is as follows:*

*"(that is, DOC, DO, ammonium, and nitrate) from the domain in steady state conditions. Thus, while the chemical species entering the domain at the inlet were consumed at varying rates by the microbial species present in the system, the rate of consumption was constant in time in each domain in all flow regimes."*

*Since we realized that spatial heterogeneity primarily resulted in reduced breakthrough times, we wanted to check if the changing breakthrough time is the lone driver for the reduction in removal of chemical species from the systems (proposed by Sanz Prat et al. (2015, 2016) as well), effectively reducing the problem to a zero dimensional (concentration changing in the time domain alone). So we used the analytical solution for first and zeroth order rate expressions to evaluate the reduction in removal with respect to reducing breakthrough time alone. However, we adapted the analytical solutions using the Damköhler number to generalize this discussion further (in Equation 5). We used a wide range of reducing breakthrough times (normalized by that in base cases, from 10% to 90% of breakthrough time in the base case) to solve Equations 5 and 6 (subsequently plotted in Fig. S8.*

**Technical Corrections:**
Lines 58-60: awkward wording, suggest rephrasing

*Thank you for the feedback. We will rephrase the sentence as follows:*
*"Microbial communities play a key role in these biogeochemical cycles since they mediate nearly all the naturally occurring processes that contribute to these cycles."*

Lines 76-78: consider simplifying to "…representative of a system's chemical and biological species, and second…representative of a system's flow and transport pathways."

*Thank you for the feedback. We will rephrase the sentence as follows:*
*"First, the reaction network should be representative of a systems' chemical and biological species, and second, the flow component of the model should be representative of a system's flow and transport pathways."*

Lines 102-103: suggest grouping citations at the end of the sentence

*Noted and addressed.*

Line 153: e.g. seems out of place, consider deleting

*Noted and addressed.*

Line 155: aerobic should be all lowercase

*Noted and addressed.*

Line 162: necromass is one word, not two

*Noted and addressed.*

Line 163: complete the second half of the sentence by describing how microbes become immobilized (biofilms etc.)

*Mathematically, the attachment of microbes depends on only the carrying capacity and concentration of immobile microbes (see sections A.3.3, L705 onwards). In a real system, we agree that immobilisation of microbes may be due to several reasons such as biofilms, interaction with the matrix etc. We rephrase the sentence as follows:*

*"Furthermore, the reaction network accounts for microbial attachment, in case of hospitable conditions, and detachment due to inhospitable conditions or velocity of the water (see section A.3.3). The detached mobile bacteria are transported by the flowing water."*

Table 1: "longitudinal" is misspelled

*Addressed.*

Line 270: this is "the" same (add "the" to the sentence)

*Addressed.*

Line 280: switch scale and spatial in the sentence

*Here, we intended to refer to the scale of the spatial scale sample. We will instead rephrase the sentence as follows for better readability:*
*"We explore flux-averaged concentrations of mobile species and spatially averaged concentrations of immobile species in 1-D, along the predominant flow direction, and explore the 2-D concentration heat*

*maps of the domain to compare the information lost when neglecting spatial heterogeneity at scales smaller than that of the sample.”*

Figure 2: Is a title necessary? The figure caption should cover the topic of the figure.

*We agree, and we will remove the title in the revised manuscript.*

Line 424: unbold sentence

*Addressed.*

Figure 4: Why is there a border around this figure?

*This is a legacy error. We will remove the border in the revised figure.*

Line 447: change to, “Since nutrient dynamics are..."

*Addressed*

Line 482: Is there a difference between dormant and inactive? The term “dormant” is used sparingly within the manuscript, so I'd suggest sticking to inactive and defining that this pool includes dormant microbes to avoid confusion.

*We agree that we should be consistent with the terminology. We will use “inactive” consistently throughout the revised manuscript.*

Lines 487-489: clunky sentence, suggest rewording

*Rephrased in the revised manuscript:*
*“The system may respond similarly to temporal fluctuations in groundwater velocities resulting from seasonal cycles as well.”*

Figure 6: It's a bit unorthodox to present figures within the discussion section. Why didn't you introduce it within the results section and then reference it in the discussion?

*We agree with the reviewer in that it is better to introduce Figure 6 in the Results section. We will rearrange the manuscript to accommodate this.*

Line 618: Change to “The regression model links the..."

*Addressed.*

Line 631: Change to “… was considered when evaluating…” (delete “for”)

*Addressed*

***References***

*Desbarats, A. J., and Bachu, S.: Geostatistical analysis of aquifer heterogeneity from the core scale to the basin scale: A case study, Water Resources Research, 30, 673-684, 1994.*

*de Marsily, G., Delay, F., Gonçalvès, J., Renard, P., Teles, V., and Violette, S.: Dealing with spatial heterogeneity, Hydrogeology Journal, 13, 161-183, 10.1007/s10040-004-0432-3, 2005.*

*Pittroff, M., Frei, S., and Gilfedder, B. S.: Quantifying nitrate and oxygen reduction rates in the hyporheic zone using 222Rn to upscale biogeochemical turnover in rivers, Water Resources Research, 53, 563-579, https://doi.org/10.1002/2016WR018917, 2017.*

Sanz-Prat, A., Lu, C., Finkel, M., and Cirpka, O. A.: On the validity of travel-time based nonlinear bioreactive transport models in steady-state flow, J Contam Hydrol, 175-176, 26-43, 10.1016/j.jconhyd.2015.02.003, 2015.

Sanz-Prat, A., Lu, C., Finkel, M., and Cirpka, O. A.: Using travel times to simulate multi-dimensional bioreactive transport in time-periodic flows, J Contam Hydrol, 187, 1-17, 10.1016/j.jconhyd.2016.01.005, 2016.

Turcke, M. A., and Kueper, B. H.: Geostatistical analysis of the Borden aquifer hydraulic conductivity field, Journal of Hydrology, 178, 223-240, 1996.

Welhan, J. A., and Reed, M. F.: Geostatistical analysis of regional hydraulic conductivity variations in the Snake River Plain aquifer, eastern Idaho, GSA Bulletin, 109, 855-868, 1997.

---

## Author Response (AR1)

**Final response to all reviewers**

*We thank all the reviewers for the constructive feedback which helped us improve the discussion of the results and overall readability of the manuscript. While we have already responded to the specific comments from the reviewers in our individual responses, we provide locations of the major changes in the revised marked up manuscript below. Reviewer comments are preceded with "RC" and our response is the italicised text following the reviewer comment.*

*A revised description of the AIC criterion and the derivation of Damköhler estimates is now included in the revision P10:L272 to P11:L301.*

RC: P1-L18: 'undertook'
*P1-L19: Replaced with 'used'.*

RC: P1-L21: In biology we have a clear nomenclature. Conditions are either 'oxic' or 'anoxic', Organisms and processes are 'aerobic' or 'anaerobic'. I suggest to use this nomenclature concisely throughout the MS.
*Changes made at several locations throughout the manuscript including P1-22*

RC: P2-L37: Here and at many other spots
RC: P2-L49: Papers of potential interest for the authors: Zhou et al. (2012) FEMS Microbiol. Ecol. 81: 230-242, Hofmann et al. (2020) Front. Microbiol. 11: 543567
RC: P2-L50: Papers of potential interest for the authors: McGuire et al. (2000) Chem Geol 169: 471–485, McGuire et al. (2005) Ground Water 43: 518–530

*P2-L50 onwards: References added at several locations in the Introduction section.*

RC: L58: "in this microbial ecosystems...": what does "in this" refer to?
Lines 58-60: awkward wording, suggest rephrasing
*P2: L60*

RC: In the Introduction section important issues such as the discrimination between 'active' and 'inactive' as well as 'mobile' and 'immobile' cells are not picked out as central points.
*P2: L68-75.*

RC: Lines 76-78: consider simplifying to "…representative of a system's chemical and biological species, and second…representative of a system's flow and transport pathways."

*P3:L84-87*

RC: L78: "Sufficiently well" for what?
*P3: L88*

RC: L81: Please add citations for the statement on biogeochemical reaction networks.
*P3: L90*

RC: L81-82: It is not clear to me how the sentence starting with "Working with ..." fits into the line of arguments here.
*P3: L91*

RC: L83: "group" instead of "groups"
*P3:L94.*

50    RC: L115: "attempted" instead of "attempt"
      *P4: L126*

      RC: L123: "Disentangle" from what? I think describe/define would be better.
      *P5: L135*
55
      RC: As highlighted in the first paragraph, the conceptual model is simplistic. While the focus
      is to test for spatial heterogeneity and flow velocity as steering factors with respect to
      carbon, nutrient and microbial dynamics, frame conditions for the model simulations are
      non-dynamic with steady-state flow and constant inflow concentrations of dissolved
60    species. In this respect, I would expect an in-depth discussion of the model output. Are
      the set frame conditions sensitive factors? How may the results change with respect to
      transient input concentrations of carbon and nutrients.
      *P5-L137,L142,L160*
      *P22-L553-572*
65    *P28-L760-763*

      RC: The authors mention the 'use of geochemical and geomicrobial observations from a
      common study site' as basis of the conceptual model. However, I could not find any
      sources (papers cited) with respect to 'values'. The concentrations of TOC, DOC, NH4,
70    NO3, O2, prokaryotic cells (active and inactive) have been selected and based on which
      studies and sites.).
      *P5-L160-162*

      RC: Line 163: complete the second half of the sentence by describing how microbes become
75    immobilized (biofilms etc.)

      *P6:L183 onwards*

      RC: In P6-L182 you say: 'The concentrations of the reactive species mimicked conditions
      observed in the subject site'. However, in the discussion it is mentioned that there were two
80    orders of magnitude difference in prokaryotic cell numbers. I ask the authors to carefully
      consider input values. Is it true that a prokaryotic concentration of mobile prokaryotic cells
      in groundwater of 10^9 have been found in the field? Seems very high to me.
      *P23: L582-598*

85    RC: Line 195: What is the correlation length used in the simulations? I couldn't find it in the
      tables.

      *P8:L218-220.*

      RC: L196: "established": can you add references for this?
90    *P8: L221-224*

      RC: L196: What are variance and anisotropy values used for the base case? When I first
      looked at Figs. S1 and S2, I was confused because the homogeneous base case was
      shown in all variance:anisotropy ratios.
95    *P9: L240*

      RC: Lines 196-198: The outcomes may be sensitive to the assumption of a second-order
      stationary random field with horizontal anisotropy. Other types of heterogeneity (e.g.,

multipoint statistical models, geometric models, or depositional process models) could lead to different (and probably even more striking) conclusions. I don't view this as a flaw of the study, since this assumption is conventional, but the assumption and its potential implications should perhaps be discussed.

*P22:L550-552.*

RC: I know that it is hard to collect reliable information from the literature with respect to microbial features in shallow aquifers. Having this in mind, one need to carefully select values for 'rate constants', 'yield coefficients', … The values summarized in Table A.4.1 originate from field studies and lab studies very different in nature, i.e. values derived from lab experiments with pure bacterial cultures. Are the chosen values sufficiently representative for the Critical Zone in Hainich and shallow aquifers in general? This at least needs to be critically discussed.
*P22: L553-565*

RC: With respect to DOC, an contstant input concentration of 800µM has been chosen. DOC degradation in soil and in groundwater is determined not only by its concentration but more likely by its quality (degradability). Has this been considered.
There is dynamics in many aspects, including flow velocity, water retention time, activity and biomass of microbes, DOM concentration and transformation, N transformation, …
Only a subset of parameters, i.e. spatial heterogeneity and flow velocity (related to residence time) has been tested. This needs to be clearly mentioned already in the Introduction section.
*P5: L137-139*

RC: Lines 245-246: Clarification is needed here to elucidate what is meant by "fit of the model" in defining the AIC criterion. It isn't clear whether this is fit to actual field observations (there are some discussed in the paper but they are not described in detail) or fit to analytical solutions (e.g. Figure S8), or something else.

*P10:L273-278*

RC: Line 280: switch scale and spatial in the sentence

*P12:L328-329*

RC: P12-L321: Is there any evidence that the portions of active and inactive cells/species are
realistic? In particular when these ratios are calculated for individual physiological guilds (nitrate reducers, ammonium oxidizers, …). See also table 4.
*P23:L594-596*

RC: L391-392: The bars in Fig. S6 are not linked to redox conditions- is it possible to do so?
*Figure updated.*

RC: P16-L411: Provide a citation that supports this statement.
*P18-L472*

RC: L444: What is AIC and what do these values mean?
*P10: L273-278*

RC: L446: Seems like a repetition of L 443f.
*P20: L503-515*

150
RC: P19-L459: The 'available' process knowledge, does it refer to the Hainich study site?
*P21-L537-538*

RC: L461: I think it would be beneficial to explicitly state the range of the scenarios.
155 *P21: L540-541*

RC: P19-L474: from carbon concentration and carbon content per cell one will not end up with gene copies per volume but cells per volume.
*P22-L576*
160
RC: Lines 487-489: clunky sentence, suggest rewording

*P23:L607-609*

RC: P20-L490 & L516: There is techniques and reports available on high-resolution
165 sampling in aquifers. The ready should not get the impression one cannot get spatially more resolved in sampling. E.g. Ronen et al. 1987 J. Hydrol. 92, 173–178, Báez-Cazull et al. 2007 Appl. Geochem. 22, 2664–2683, Smith et al. 1991Contam. Hydrol. 7, 285–300, Anneser et al. 2008 Appl Geochem 23:1715–1730.
*P24:L641-645*
170
RC: P20-Fig. 6: What do you mean with 'oxic cells'. Please change.
*P17: Figure 3*

RC: P21-L528: Consider the review paper of Smith et al. 2018 FEMS Microb. Ecol. 94:
175 fiy191
*P25: 651-654.*

RC: L553-555: A verb is missing in this sentence.
*P25:L679*
180
RC: P22-L565: I fully agree with this statement. In many cases the contribution of the obile fraction of microbes can be neglected in terms of 'transformation processes'. Findings from other studies (like the one already cited Grösbacher et al. 2018) are not discussed in comparison to the model outcome.
185 *P23: L582-594*
*P26: L689-692*

RC: L627: Can the authors elaborate on the significance of this results for environmental systems? For example, when and where do they expect these heterogeneities to be most
190 significant?
*P28: L754-763*

RC: P24-L630: mention at which spatial scale.
*P28: L766*
195

---

## Author Response (AR2)

**Authors'Response**

**25. November 2021**

*We thank all the reviewers once again for the constructive feedback which helped us improve the discussion of the results and overall readability of the manuscript. We have taken careful note of the*
5 *technical corrections and implemented them in the revised manuscript. The locations of these corrections are detailed below next to the suggested technical corrections. Reviewer comments are preceded with "RC" and our response is the italicised text following the reviewer comment. Upon proof-reading the manuscript, we also noted several additional locations where we deemed modification of phrasing to be appropriate considering consistency of language/wording, ease of readability, and referencing. These*
10 *modifications are noted separately at the end of this response.*

RC: The section in L754-763 illustrates in which systems spatial heterogeneity is important. It is not entirely clear to me why the authors compared shallow systems to deep subsurface systems here- did they want to state that spatial heterogeneity is less important in deep subsurface systems? If so, it
15 would help to state this explicitly in the text.

*The intention of these lines was to suggest physical systems where spatial heterogeneity may play a role, keeping in mind our description of dominant flow processes. This is why we preceded the examination of shallow and deep subsurface systems by the description of the dominant flow process (P27: L727-"advection dominated system ..."). Keeping in mind the reviewer's suggestion, we have added a*
20 *clarification that these systems are mere examples:*

*L730-732- "... In contrast, in the deep subsurface microbial activity is lower and rather relies more often on inputs from the matrix material, which is ubiquitous and does not rely on transport for access to nutrients or energy gradients. It must be noted that this generic description of dominant processes serves to give examples for reactive systems for the purpose of our discussion and may vary from site-to-site depending*
25 *on specific site characteristics."*

**Technical corrections:**

RC L35: Do the authors mean terminal electron acceptors?

*L52- Corrected*

30 RC L52: I think there should be an "and" between the two citations.

*L51- Corrected*

RC L72: The authors could use "account for" instead of "vary between".

*L70- Corrected*

RC L112: "several studies" instead of "several studied"

35 *L108- Corrected*

RC L146: "effectively upscaling" instead of "effective upscaling" (or, alternatively: "effective upscaling of these")

*L141- Corrected*

RC L160: "a" instead of "and"

40 *L154- Corrected*

RC L267: "in addition to these dissolved" instead of "additionally, to these dissolved"

*L260- Corrected*

RC L509: delete ")"

*L492 - Corrected*

RC L645: I don't understand "though sub-sampling scale heterogeneities will not be resolved". Maybe there is a missing word?

*L614-615- We realized that the phrasing was quite complex, so we have rephrased as follows:*

*"even though heterogeneities at scales lower than that resolved by the sampling scheme will remain unobserved."*

RC L659: Do the authors mean that the relative composition is similar in the cited studies and herein?

*L628-629- We meant that the representation of the functional groups in the immobile fraction is similar to that in the mobile fraction. Considering the ambiguity induced, we rephrased the sentence as follows:*

*"Our results suggest that the relative composition of species in the mobile and immobile subcommunities is however similar…"*

RC L757: "receives" instead of "received"

*L727- Corrected*

RC L759: "does not" instead of "doesn't"

*L729- Corrected.*

**Authors' corrections:**

*L11- Updated email address of the corresponding author.*

*L98 – "settings" instead of "setting"*

*L114- "depending on the sampling resolution" instead of "depending on sampling resolution"*

*L173- "The reaction network also accounts for…" instead of "…, and as well as for the…"*

*L174- ", lumping all…" instead of "The latter lumps all…"*

*L188 – "Hainich CZE" instead of "Hainich Critical Zone Exploratory"*

*L200, L211, L229 – "Jing et al., 2018" instead of "Jing et al., 2017"*

*L212-:L216- The references added during the review stage were missing from the bibliography. We have now corrected this issue.*

*L274 – "…ratio of the transport time scale…" instead of "…ratio of the advective transport time scale…"*

*L332- "DO" instead of "oxygen"*

*L378 – "…field and increase in…" instead of "field, and increasing …"*

*L527- "… it captures varying  microbial respiration and growth regimes…"instead of "… it captures varying respiration and microbial regimes…"*

*L540 – "This compensates for our approach" instead of "This approach compensates for our approach"*

*L729- "… relies more often on inputs… " instead of "… relies on inputs …"*